# Uncertainties in the Seismic Assessment of Historical Masonry Buildings

**Igor Tomić** [1], **Francesco Vanin** [1,2] and **Katrin Beyer** [1,*]

1   École Polytechnique Fédérale de Lausanne (EPFL), School of Architecture, Civil and Environmental Engineering (ENAC), Earthquake Engineering and Structural Dynamics Laboratory (EESD), 1015 Lausanne, Switzerland; igor.tomic@epfl.ch (I.T.); francesco.vanin@epfl.ch (F.V.)
2   Résonance Ingénieurs-Conseils SA, 1227 Carouge, Switzerland
*   Correspondence: katrin.beyer@epfl.ch

**Abstract:** Seismic assessments of historical masonry buildings are affected by several sources of epistemic uncertainty. These are mainly the material and the modelling parameters and the displacement capacity of the elements. Additional sources of uncertainty lie in the non-linear connections, such as wall-to-wall and floor-to-wall connections. Latin Hypercube Sampling was performed to create 400 sets of 11 material and modelling parameters. The proposed approach is applied to historical stone masonry buildings with timber floors, which are modelled by an equivalent frame approach using a newly developed macroelement accounting for both in-plane and out-of-plane failure. Each building is modelled first with out-of-plane behaviour enabled and non-linear connections, and then with out-of-plane behaviour disabled and rigid connections. For each model and set of parameters, incremental dynamic analyses are performed until building failure and seismic fragility curves derived. The key material and modelling parameters influencing the performance of the buildings are determined based on the peak ground acceleration at failure, type of failure and failure location. This study finds that the predicted PGA at failure and the failure mode and location is as sensitive to the properties of the non-linear connections as to the material and displacement capacity parameters, indicating that analyses must account for this uncertainty to accurately assess the in-plane and out-of-plane failure modes of historical masonry buildings. It also shows that modelling the out-of-plane behaviour produces a significant impact on the seismic fragility curves.

**Keywords:** stone masonry; seismic assessment; equivalent frame models; uncertainty analysis; Latin hypercube sampling; incremental dynamic analysis





## 1. Introduction

Historical masonry buildings can be highly vulnerable to earthquake damage, as has unfortunately become evident in several historical city centres of Europe. An example of the damage is shown in Figure 1. To decrease this vulnerability, it is necessary to adopt assessment procedures that accurately reflect all the peculiarities of historical masonry. Unfortunately, correct evaluations of seismic performance are often hindered by a lack of information regarding the materials, structure and connections. To complicate this further, the heterogeneity of the construction material and structural detail often increases as buildings degrade over their life span or as a result of interventions and alternations. At the same time, the extensive testing or measuring of the properties is often not feasible due to either the high cost of experimental campaigns or limitations imposed due to the cultural value and protected status of a building. Last, but not least, connections between the structural elements, such as floor-to-wall interfaces and wall-to-wall interlocking have properties that are difficult to predict.

One approach to consider all these uncertainties is to use field data as basis for empirical vulnerability curves. This approach has been applied by Zuccaro et al. for Italian masonry buildings [1] and by Ruggieri et al. for masonry one-nave churches that were

subjected to the Valle Scrivia Earthquake, 2003, Piemonte, Italy [2]. If numerical models are used for deriving the vulnerability curves, the models must adequately describe the geometry, morphology, connections and support conditions and must balance this with maintaining cost-effective computations that can be applied outside of the academic domain in seismic risk areas worldwide. To face these challenges, different modelling techniques and approaches have been developed to simulate the behaviour of masonry, and these techniques vary greatly in complexity levels and associated computational burden [3,4]. No matter which model is used, the question still remains on the choice of material and modelling parameters.

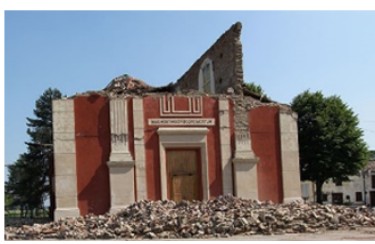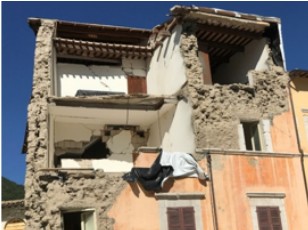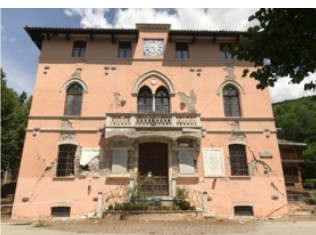

**Figure 1.** Example of masonry buildings damaged in recent earthquakes in Italy.

When dealing with uncertainty, it is common to differentiate between aleatory randomness and epistemic uncertainty. Aleatory randomness is the natural randomness in the process, and the epistemic uncertainty stems from limited data and knowledge [5,6]. Different approaches with different levels of complexity were applied by several authors to treat these uncertainties in modelling historical masonry buildings.

For example, many Incremental Dynamic Analyses (IDAs) were performed by Dolsek [7] to deal with both the aleatory randomness stemming from differences in seismic records and the epistemic uncertainty stemming from material and modelling properties. Latin Hypercube Sampling (LHS) was used to distribute the material and modelling properties. The effect of uncertainty on the seismic response parameters was presented in terms of summarized IDA curves and dispersion measurements. The study concluded that epistemic uncertainty did not significantly affect the seismic response parameters in the range farther from collapse, but that the median collapse capacity was reduced when the epistemic uncertainties had been accounted for.

Rota et al. [8] dealt with the issues of the knowledge levels and associated values of a confidence coefficient in the assessment method for existing buildings included in Eurocode 8, Part 3 [9]. The proposed methodology assessed epistemic uncertainties using so-called variability factors, which were calibrated based on a logic-tree approach and aimed to represent the dispersion of results due to each of the uncertainties. This resulted in a global safety coefficient, applied directly to the structural capacity.

Saloustros et al. [10] performed a stochastic analysis based on a Monte Carlo simulation to investigate the effect of material mechanical parameters on the evaluation of seismic fragility. The methodology was applied to the case study of the Santa Maria del Mar church in Barcelona and illustrated the sensitivity of the seismic analysis to the uncertainty in the material properties, as two different collapse mechanisms were possible depending on the input parameters. Finally, analytical seismic fragility curves were proposed by considering several sources of uncertainty regarding the material properties.

Vanin et al. [11] instead proposed the logic tree approach as an effective method aimed at replacing the Monte Carlo simulations, which can be computationally too expensive for practical use. Optimal sampling points were defined using a moment-matching technique, and a combination of weights were applied to the branches. As a more refined and a novel approach in structural engineering, a Point Estimate Method was adopted to further reduce the number of the required non-linear simulations. A case study historical masonry building modelled by the equivalent frame model (EFM) approach was used to test both methods, which both showed good accuracy when compared with the Monte

Carlo method. The in-plane displacement capacities of piers was identified as a major source of uncertainty.

This study strove to capture the key mechanical parameters of masonry on an elemental level using appropriate material laws while maintaining a simplicity that would allow for multiple dynamic analyses to be performed. This is necessary to account for the aleatory and epistemic uncertainties involved in the modelling of masonry buildings. To this scope, EFMs were used, as they fulfilled the necessary requirements and are one of the most widely applied engineering tools for the seismic analysis of unreinforced masonry buildings for both scientists and practitioners alike. For the EFM, the macro-element by Vanin et al. was used [12]. This macro-element cannot only capture the in-plane behaviour of masonry elements but also their out-of-plane behaviour. This feature addressed the weakness of many EFM formulations, which require a separate analysis to account for out-of-plane behaviour, most commonly through separate limit analysis. Furthermore, this modification provided a framework to analyse the impact of mixed in-plane and out-of-plane failure modes.

In the following sections, the case study buildings and details of the modelling approach are first described. Then, material and modelling properties with their respective normal and lognormal distributions and the earthquake record used in the incremental dynamic analyses are presented and discussed. This is followed by the description of the methodology for the statistical evaluation of the data. The results for each case study building are presented and discussed. For each building two sets of analyses are performed: first, the building is modelled with the out-of-plane capability of the macroelement enabled and with non-linear wall-to-wall and floor-to-wall connections, and then without out-of-plane capability of the macroelement and rigid connections. Finally, conclusions on uncertainties in modelling unreinforced masonry buildings are drawn.

## 2. Case Studies

Two buildings were selected as case studies. The two typologies represent typical structures often found in city centres, which include (i) stiff monumental buildings, and (ii) tall and slender masonry buildings.

### 2.1. Holsteiner Hof

Holsteiner Hof, shown in Figure 2, is a historical stone masonry building located in the city centre of Basel. Built in 1752, it is a landmark of cultural heritage and is considered a typical representative of buildings built during the 17–19th century. The building has two storeys with a height of 4.50 m and a rectangular floor plan ($26 \times 14$ m$^2$). The wall thickness of both storeys is 60 cm, while the thickness of the spandrels is only 30 cm. The gables are triangular walls with a thickness of 45 cm. The floors are composed of timber beams spanning the shorter direction and one layer of planks nailed to the beams. The timber beams are simply supported on the walls and horizontal forces are transferred as friction forces. The roof system is a truss that was retrofitted with a secondary structure. Minor retrofitting interventions were performed in 1976–1979 but did not alter the structure in a significant way.

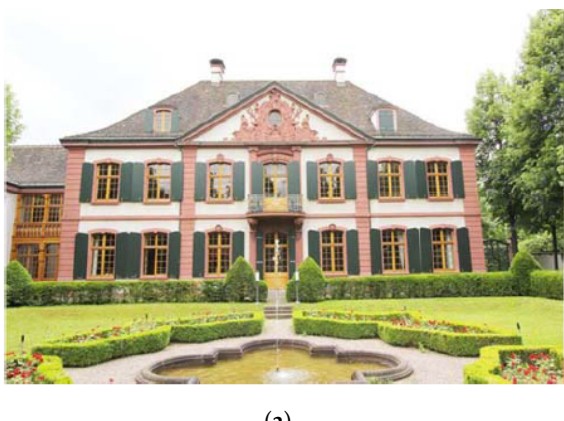
(a)

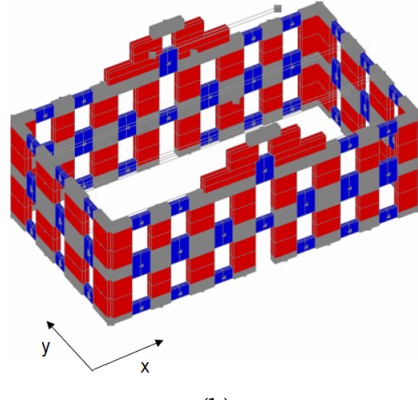
(b)

**Figure 2.** Holsteiner Hof. (**a**) Main façade. (**b**) Numerical model with the coordinate system.

### 2.2. Lausanne Malley

Lausanne Malley, shown in Figure 3, is a representative example of a tall and slender unreinforced stone masonry residential building in Lausanne. It was constructed in the second half of the 19th century with a rectangular floor plan (14 × 12 m²) and a regular layout with reinforced concrete footings under the walls. The wall thickness varies from 60 to 25 cm along with the height of the building. It has six storeys, and the storey height varies between 2.80–3.20 m. The original timber floors are composed of timber beams spanning the shorter direction and one layer of planks nailed to the beams. As for the Holsteiner Hof, the timber beams are simply supported on the walls and horizontal forces are transferred as friction forces. The roof has the original wooden truss structure. Sound-proof retrofitting has been recently performed [13], which slightly improved the seismic capacity of the structure.

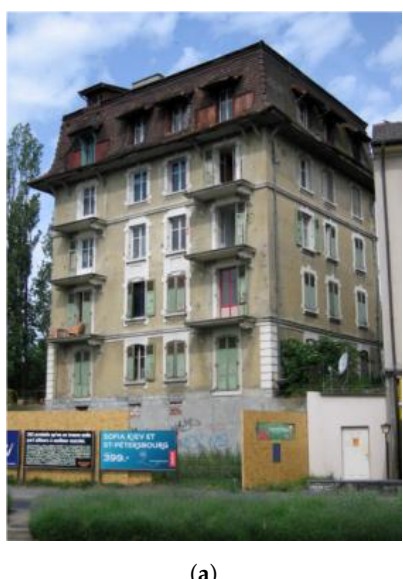
(a)

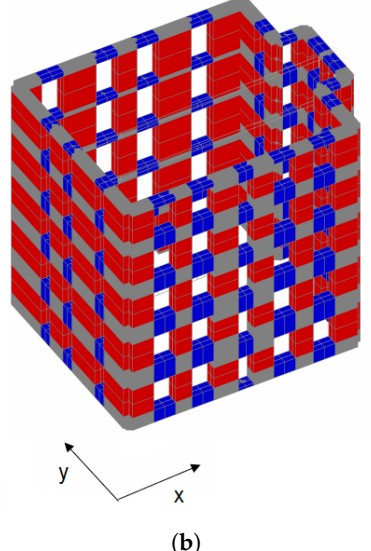
(b)

**Figure 3.** Lausanne Malley. (**a**) Main façade. (**b**) Numerical model with the coordinate system.

### 2.3. Modelling Approach

The buildings were modelled using the equivalent frame model (EFM) approach. EFM works by discretizing façades into piers, spandrels and rigid nodes using beams or macroelements [14]. For this work in particular, a macroelement newly developed by Vanin et al. [12] was used that builds on the work of Penna et al. [15] to precisely model the behaviour of masonry panels in-plane. The macroelement is formulated as

a one-dimensional element defined by three nodes in three-dimensional space. Three non-linear sections account for axial deformation, and the central section accounts for the non-linear shear response. The out-of-plane response is coupled with the in-plane response. Nonlinear geometrical effects are captured through a second-order Taylor series expansion of the compatibility equations, obtaining the $P - \Delta$ formulation. To correctly simulate the out-of-plane behaviour of a masonry building, it is not sufficient to choose a macroelement formulation that captures the out-of-plane behaviour, if walls are restrained in the numerical model by rigid wall-to-wall and floor-to-wall connections. Therefore, it is crucial that the model allows for relative displacements between the nodes of the intersection walls and between the nodes of the floor and the wall [16]. For this purpose, interface elements and material models for wall-to-wall and wall-to-floor connections were implemented in OpenSEES [17].

All piers and spandrels were assigned the same material properties, while the nodal regions between them were modelled as rigid. Timber floors were modelled as an orthotropic elastic membrane, with a higher stiffness in the direction of the beam span and a lower stiffness in the orthogonal direction. The shear stiffness of the membrane was defined by the shear modulus, meaning the membrane was defined by the two moduli of elasticity in two orthogonal directions, as well as the shear modulus and the thickness of the diaphragm. The floors were modelled as linear elastic, but the connection between the floor and the wall was modelled to account for a nonlinear behaviour and a possible connection failure, which can result in the out-of-plane failure of a pier. The nodes of the floor were modelled separately from the nodes of the walls to allow the possible relative displacement. This connection was modelled using a zero-length element to which a frictional model was assigned. The sliding occurred in the direction perpendicular to the wall, and pounding occurred in the opposite direction. Although it is possible to model slip parallel to the wall, for this study, the connection parallel with the wall was assumed to be linear elastic. Therefore, the properties of the zero-length element were defined by the friction coefficient characterising the interface between floor beams and walls, the modulus of elasticity and the shear modulus of this connection [16]. Another set of zero-length elements was used to model the connection between orthogonal walls. This connection simulated the potential formation of a vertical crack and separation of the orthogonal walls due to poor interlocking, which could lead to the out-of-plane failure of a wall. The one-dimensional material for the interface was defined as a damage tension law with exponential softening and a linear elastic model in compression [16]. The material was defined by the elastic modulus, the tensile strength and the Mode I fracture energy.

*2.4. Material and Modelling Parameters*

A total of 11 material and modelling parameters were selected to perform the uncertainty analysis. The material parameters of masonry were the following: $E_m$, modulus of elasticity; $G_m$, shear modulus; $f_{cm}$, compressive strength; $\mu_m$ and $c_m$, friction coefficient and cohesion of masonry, respectively. No in situ tests had been performed. For this reason, median values were chosen based on the experimental campaigns performed on masonry of a similar typology as described previously [18–21] and were applied to all piers and spandrels, i.e., any spatial variability in masonry material properties was not considered. The floor stiffness factor ($k_{floor}$) multiplied the default stiffness values of the flexible floor diaphragm, which were computed as: $E_1 = 10$ GPa, $E_2 = 0.5$ GPa, and $G = 10$ MPa according to previous work of Brignola et al. [22,23]. The wall-to-wall connection factor $f_w$ multiplied the default wall-to-wall connection strength. This default wall-to-wall connection strength was calculated according to Fontana et al. [24]. The floor-to-wall friction coefficient $\mu_{f-w}$ was directly applied to the frictional floor-wall connection, with the mean value based on the work of Almeida et al. [25]. The drift capacities in flexure and shear, $\delta_{c,flexure}$ and $\delta_{c,shear}$, are the limit collapse flexural and shear drifts at which the lateral stiffness of the macroelement is set to zero, chosen according to Vanin et al. [26]. An initial stiffness and mass proportional Rayleigh damping model was applied. The damping ratio was also a

parameter that was considered uncertain. For each set of parameters, the modal properties were calculated first and then the Rayleigh damping model parameters were computed such that the damping ratios at the first and sixth mode corresponded to the damping ratio of this set of parameters. All parameters were assigned normal or lognormal distributions for the LHS, as shown in Table 1.

**Table 1.** Assumed distributions of material and modelling parameters.

| Parameter | Unit | Distribution | Mean | Lognormal Mean | Standard Deviation | Reference |
|---|---|---|---|---|---|---|
| $E_m$ | Pa | Normal | $3.5 \times 10^9$ | - | $1.0 \times 10^9$ | [18,20] |
| $f_{cm}$ | Pa | Normal | $1.3 \times 10^6$ | - | $0.35 \times 10^6$ | [18,20] |
| $\xi$ | - | Normal | 0.02 | - | 0.005 | [16] |
| **Parameter** | **Unit** | **Distribution** | **Median** | **Lognormal Mean** | **Standard Deviation** | **Reference** |
| $G_m$ | Pa | Lognormal | $1.5 \times 10^9$ | 21.13 | 0.5 | [18,20] |
| $c_m$ | Pa | Lognormal | $0.233 \times 10^6$ | 12.36 | 0.5 | [18,20] |
| $\mu_m$ | - | Lognormal | 0.25 | $-1.39$ | 0.3 | [18,20] |
| $k_{floor}$ | - | Lognormal | 1 | 0 | 0.5 | [22,23] |
| $f_w$ | - | Lognormal | 1 | 0 | 0.3 | [24] |
| $\mu_{f-w}$ | - | Lognormal | 1 | 0 | 0.3 | [16,25] |
| $\delta_{c,flexure}$ | - | Lognormal | 0.01035 | $-4.57$ | 0.2 | [26] |
| $\delta_{c,shear}$ | - | Lognormal | 0.007 | $-4.96$ | 0.2 | [26] |

Additionally, correlation coefficients were imposed between parameters that are often correlated in experimental campaigns to avoid generating unrealistic data sets, i.e., with the lower-bound value for modulus of elasticity and the upper-bound value for the compressive strength. Imposed correlation coefficients are shown in Table 2. A strong correlation was imposed between the modulus of elasticity, shear modulus and compressive strength. A moderate correlation was imposed between collapse drift values in flexure and shear for masonry elements. It should be noted that the imposed correlation coefficients between the parameters were only estimated based on engineering judgment.

**Table 2.** Correlation coefficients between input parameters.

| Correlated Parameters | Correlation Coefficient | Strength of Correlation |
|---|---|---|
| $E_m$, $G_m$, $f_{cm}$ | 0.7 | Strong |
| $\Delta_{c,flexure}$, $\Delta_{c,shear}$ | 0.5 | Moderate |

### 2.5. Earthquake Record

A bi-directional incremental time history analysis was performed for each set of parameters. The analyses were carried out using one ground motion record. The chosen record is the Montenegro Albatros 1979 record, with a PGA of 0.18 g in the north-south direction and 0.21 g in the east-west direction, shown in Figure 4 [27]. This record was selected because of its rather broad frequency content. The initial acceleration for the Holsteiner Hof was set to 100% of the recorded PGA. The north-south direction was applied in the negative direction of the $x$-axis and the east-west direction was applied in the positive direction of the $y$-axis. Response spectra are shown in Figure 5. The initial acceleration for the Lausanne Malley building was set to 50% of the same. These values were chosen as starting points because the initial numerical analyses showed that no

failures occurred for the selected PGA levels. Afterwards, the acceleration was increased by 50% of the original record levels for the Holsteiner Hof, and by 25% of the original record for the Lausanne Malley building up to the point of failure. A smaller increment level for the Lausanne Malley building was chosen because of its greater fragility.

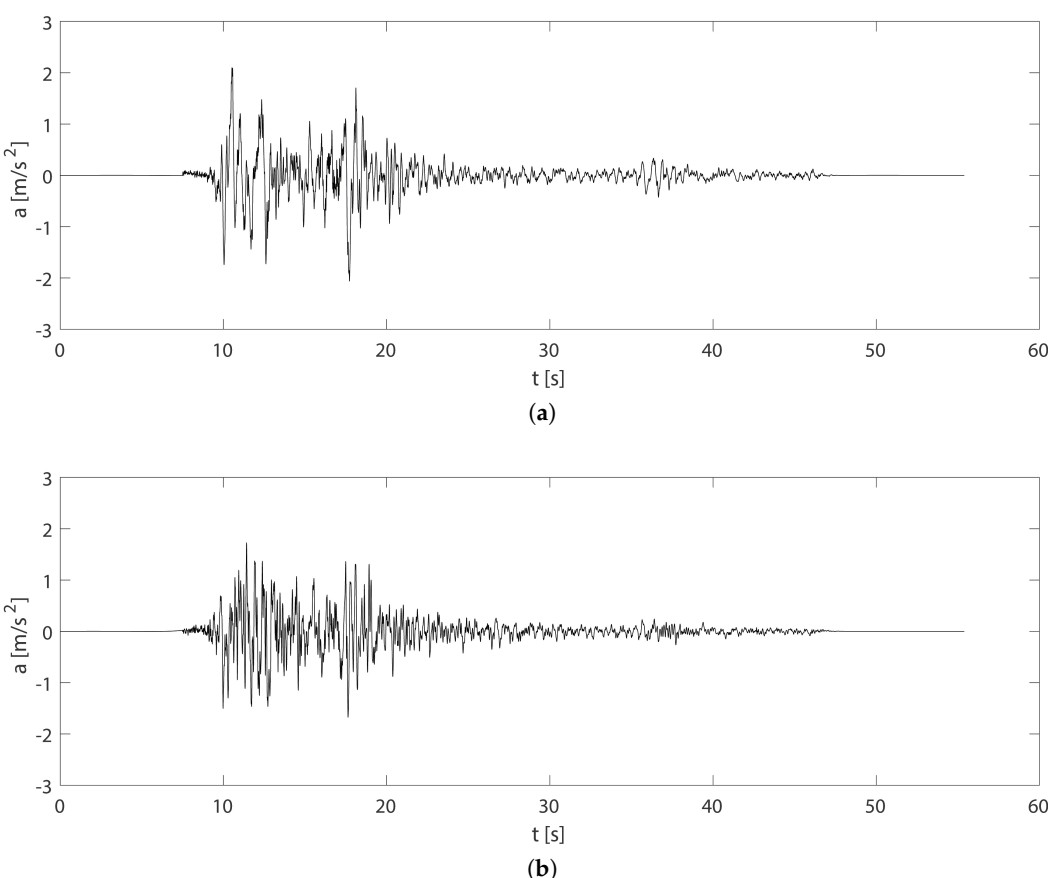

**Figure 4.** Processed acceleration time-histories of the Montenegro 1979 earthquake. Albatros station records: (**a**) east-west direction. (**b**) north-south direction [27].

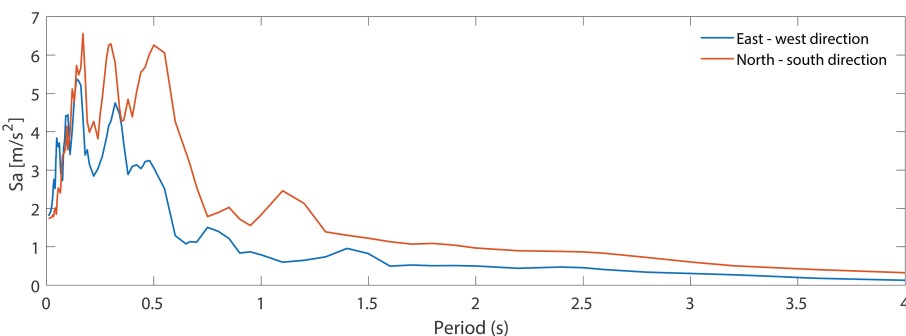

**Figure 5.** Acceleration response spectra of the Montenegro 1979 earthquake [27].

## 3. Methodology

Latin Hypercube Sampling (LHS) was performed to account for the epistemic uncertainty of material and modelling parameters. To do this, 400 sets of 11 parameters were generated in two steps. First, each parameter was assigned as a normal or lognormal distribution according to values from literature, buildings codes or experiments, as defined in the previous section. The generated $11 \times 400$ matrix contained one set of parameters in each column, which was used for a single incremental dynamic analysis. The marginal

distribution of each column was adjusted so that points were uniformly distributed on the probability scale [28]. Correlation matrices were imposed between the parameters to avoid unrealistic sample sets, e.g., a sample with a lower bound value for the modulus of elasticity and an upper bound value for the compressive strength. When all parameters were generated, the actual correlation matrix was calculated and compared to the imposed one, and this difference between the two was evaluated by the objective function that was minimized, called the norm. This methodology for minimizing the norm by performing random permutations of random elements within each vector was proposed by Dolsek et al. [7] using a method called simulated annealing that was originally proposed by Vovrechovsky et al. [29], as an approach to find the global minimum of an objective function that might feature many local minima. The norm is re-evaluated after each permutation, and it is accepted if the norm decreased or rejected if the norm increased. Once the final set of material and modelling parameters was generated, it was used to perform incremental dynamic analyses [30].

A different approach were used in the past to reduce the computational cost by limiting the number of required non-linear analyses, i.e., a novel use of a point estimate method was proposed by Vanin et al. [11]. This method is effective for more detailed analyses, such as discrete element models which are computationally demanding. Our approach relies on the low computational burden of running equivalent frame models in OpenSEES. Though this removes the requirement for keeping the number of analyses low, it is still better to avoid redundant analyses stemming from the application of the classical Monte Carlo method [31], which does not benefit the statistical output. Therefore, for this work, LHS was used for the selection of random parameters as the best balance between computational burden and accuracy.

### 3.1. Seismic Response Parameters

To evaluate seismic response parameters, each of the 400 IDA curves was first plotted, together with the mean curve and the 16th and 84th percentile curves. Second, correlations between material and modelling parameters and seismic demand parameters were evaluated for the selected PGA. The PGA was selected individually for each building to balance between avoiding too many failures while still activating the dominant mechanisms. Correlations were evaluated in terms of the linear correlation coefficients between the chosen material and modelling parameters and the maximum values of seismic response parameters. Correlation coefficients were displayed for parameters that passed the P-test with probability values of less than 5%, which present strong evidence against a null-hypothesis that two variables are not correlated. In other words, it means that a $p$-value of less than 5% represents strong evidence of a correlation. Chosen seismic response parameters are listed in Table 3.

**Table 3.** List of seismic response parameters for which the maximum absolute values are observed.

| | |
|---|---|
| BSHX | Base shear in x-direction |
| BSHY | Base shear in y-direction |
| RDX | Average roof displacement in x-direction |
| RDY | Average roof displacement in y-direction |

### 3.2. Failure Criterion

Among all the limit states, the collapse limit state is the most challenging to model because the influence of non-linear material models and modelling assumptions is large, and problems related to the numerical convergence and stability of the solution are frequent [32]. For the EFMs used in this study, the equilibrium can be lost for two reasons: (i) excessive out-of-plane deformation when the $P - \Delta$ effect causes the loss of the global equilibrium or (ii) a series of in-plane failures until the global equilibrium cannot be reached. Alternatively, the activation of reason (ii) could lead to (i). These in-plane failures were

accounted for at the element level for the macroelement developed by Vanin [12] used in this study. More specifically, when the pier reaches either the shear or flexural drift limit, the lateral stiffness and strength are set to zero, but the pier retains the ability to transfer axial load. This is illustrated in Figure 6 for the flexural and shear failures in a pier reaching drift limits set as 1.2% and 0.9% for flexure and shear, respectively.

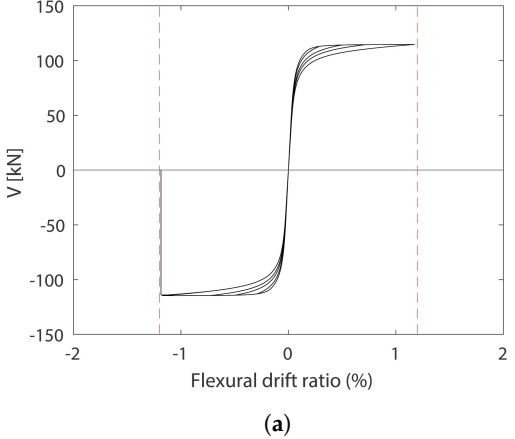

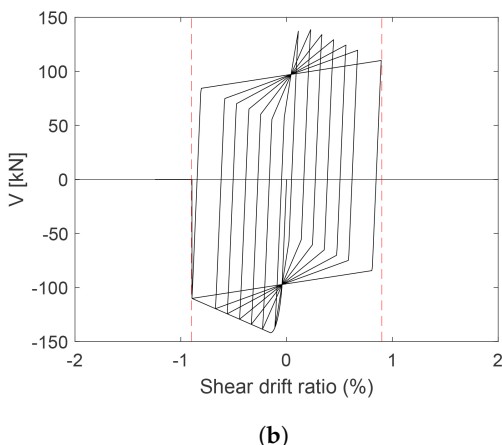

(**a**)  (**b**)

**Figure 6.** Pier lateral strength and stiffness loss after reaching a collapse drift value. (**a**) Flexural. (**b**) Shear.

In the study presented here, equilibrium loss in a particular step of an incremental dynamic analysis was considered to be a failure. Then, the failure criterion was subsequently applied to detect the cause of the loss of the global equilibrium. This applied failure criterion was inspired by one of the criteria described by Penna et al. [32]. Here, failure was defined as 50% of the piers in one direction reaching the limit drift, so the criterion was updated to consider in-plane failure when 50% of the piers in the same direction of one storey of one unit reached their drift limits. Whether it was labelled as a flexural or shear in-plane failure depended on the predominant failure mode of the piers involved. In parallel, each pier's relative out-of-plane deformation was checked. Out-of-plane rotation around the middle node of the pier was also checked, and the relative displacement between two floors was considered. This procedure was repeated for each step of the analysis that had a loss of equilibrium until one of the criteria was reached and failure of the building marked and localized, as shown in Figure 7. A final check was performed to ensure that one of the failure criteria was reached and to eliminate a potential erroneous numerical loss of equilibrium. If the equilibrium was lost but none of the failure criteria was reached, the particular IDA was discarded to eliminate potential erroneous numerical losses of equilibrium . However, this occurred in less than 0.5% of the cases.

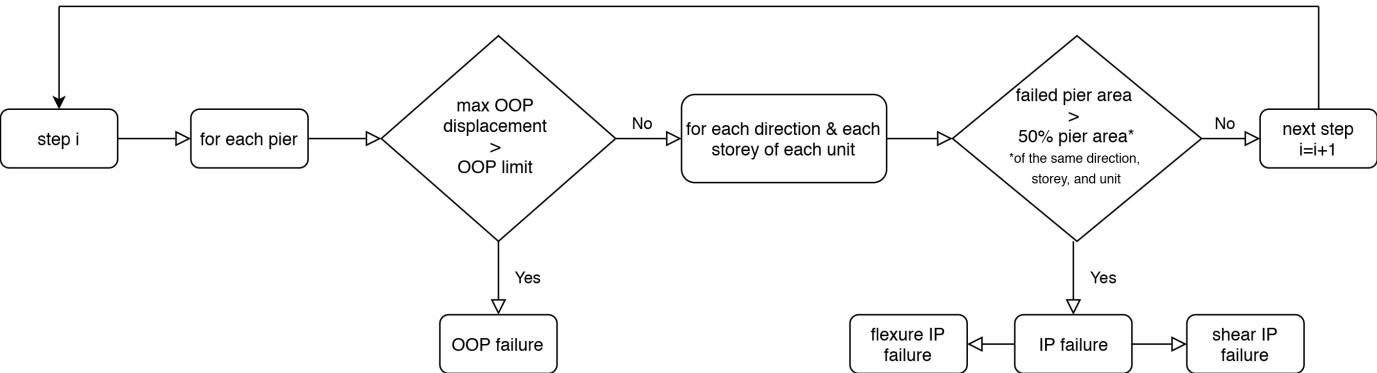

**Figure 7.** Building failure criterion chart describing the procedure to classify the building model as collapsed. OOP, out-of-plane failure; IP, in-plane failure.

The proposed methodology analysed the failure from a few different points. First, it showed the probability of failure at a certain PGA using seismic fragility curves. Then, the linear correlation coefficients were computed between the PGA at failure and the material and modelling parameters. These coefficients were then filtered using the p-value of correlation of each parameter with the PGA at failure, where only those with a p-value of less than 5% were plotted. Finally, the failure type and localization were detected for each model. Failures were divided into in-plane and out-of-plane failures, and their localization was performed according to Table 4, which uses Lausanne Malley building as an example. The same concept was applied to other buildings and updated according to the number of storeys. Overall, the methodology deals with three important aspects of the failure uncertainty: the impact of parameter uncertainty on the PGA collapse capacity of the building, collapse mechanism and collapse localisation.

**Table 4.** Lausanne Malley: Failure location divided by units, floors and directions.

| | |
|---|---|
| F1 x-dir | 1st floor in x-direction |
| F1 y-dir | 1st floor in y-direction |
| F2 x-dir | 2nd floor in x-direction |
| F2 y-dir | 2nd floor in y-direction |
| F3 x-dir | 3rd floor in x-direction |
| F3 y-dir | 3rd floor in y-direction |
| F4 x-dir | 4th floor in x-direction |
| F4 y-dir | 4th floor in y-direction |
| F5 x-dir | 5th floor in x-direction |
| F5 y-dir | 5th floor in y-direction |

## 4. Results

Herein, IDAs were performed for each of the two case-study buildings for 400 sets of material and modelling parameters. The same procedure was repeated twice for each case-study building, once including the out-of-plane capability of the macroelement and non-linear connections, and once without the out-of-plane capability of the macroelement and rigid connections. The aim of this procedure was to enhance the understanding of the impact that neglecting the out-of-plane and non-linear connections component from the equivalent frame analysis produces in terms of fragility curve. The OpenSEES models and the sets of material and modelling parameters used for the IDAs are provided in the supplementary material.

First, the seismic demand parameters were evaluated in terms of IDA curves and correlations with input parameters. Then, modal periods, seismic fragility curves and correlations of the PGA at failure with input material and modelling parameters were calculated. Finally, the statistics were evaluated regarding failure types and location.

### 4.1. Holsteiner Hof

Holsteiner Hof was modelled as described in Section 2.1. The walls were modelled using macroelements, the floors were modelled using orthotropic membrane elements, and the wall-to-wall and floor-to-wall connections were modelled using zero-length elements and non-linear material models.

#### 4.1.1. Seismic Response Parameters

For each of the 400 sets of analyses, the IDA curves are plotted in Figure 8. In this figure, the chosen output variable can be seen to evolve with the PGA increment. Since the parameter values are quite scattered, the mean curve and 16th and 84th percentile curves are plotted. Higher scatter of the roof displacement was observed for the x-direction.

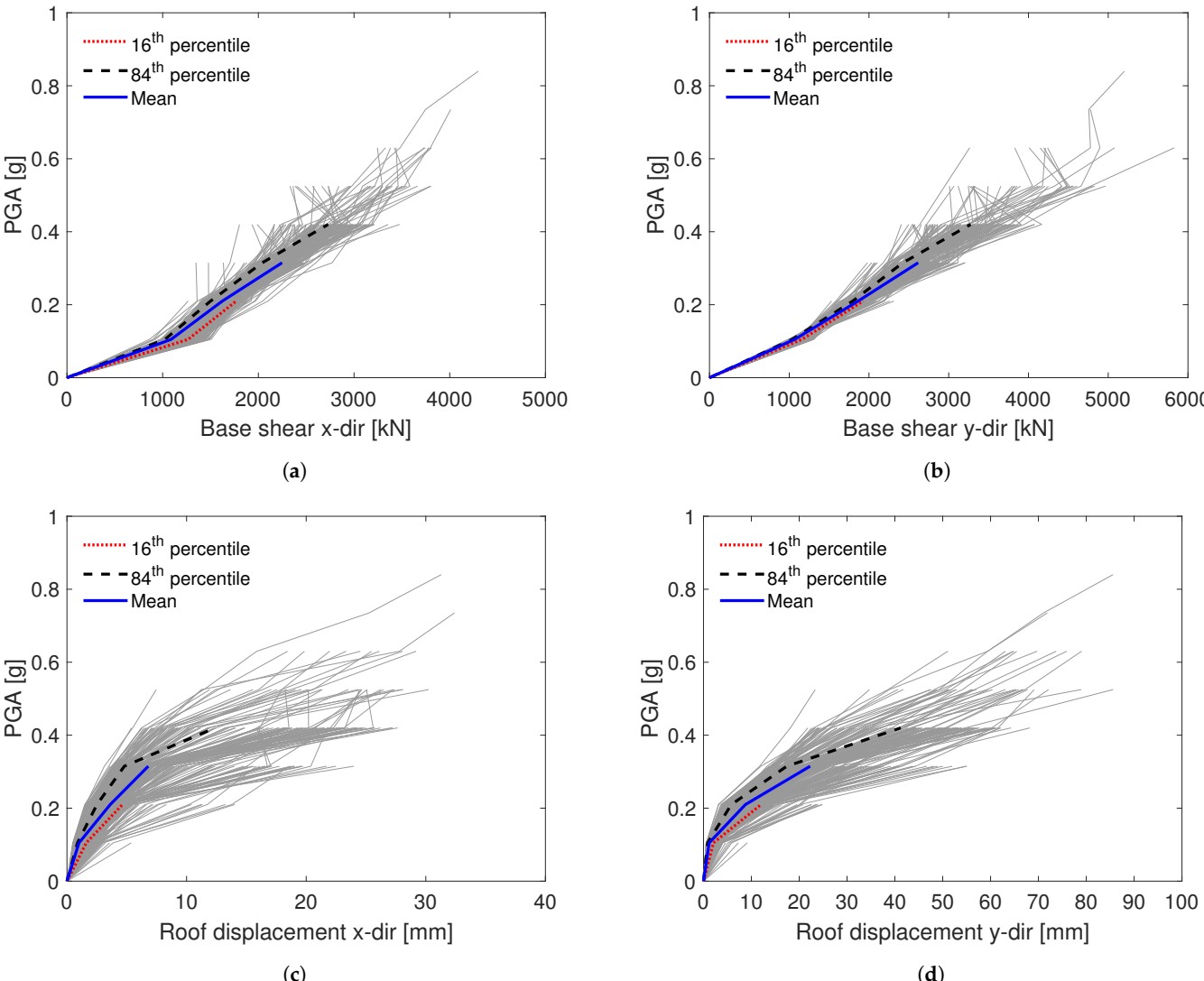

**Figure 8.** IDA curves for the Holsteiner Hof model for various seismic demand parameters: (**a**) Total base shear in x-direction. (**b**) Total base shear in y-direction. (**c**) Average roof displacement in x-direction. (**d**) Average roof displacement in y-direction.

The correlation between the base shear and roof displacement demands and the input parameters for the PGA of 0.21 g is displayed in Figure 9. This PGA value was chosen because it provided sufficiently large seismic demand parameters without triggering too many failures. For clarity, only the Spearman rank correlations for the parameters with a *p*-value of less than 5% are displayed. The strongest negative correlation was detected between the roof displacements and modulus of elasticity, shear modulus and compressive strength, followed by the Rayleigh critical damping ratio and the cohesion of masonry. The modulus of elasticity, shear modulus and compressive strength, together with the cohesion of masonry and the floor stiffness showed a positive correlation with the base shear. Since the PGA was too low to trigger element failures, in-plane drifts at collapse did not correlate with the parameters. Base shear and roof displacement demands were not correlated with the parameters of the floor-to-wall connections, suggesting that the connection capacity was not exceeded. The base shear and roof displacements were therefore dependent on the assumed floor stiffness.

### 4.1.2. Failure Analysis

Figure 10 shows the results of 400 IDAs for the Holsteiner Hof model in terms of modal periods, fragility curve and the correlations between the input parameters and the predicted PGA at failure. At the start of each IDA, modal periods were determined to calculate the Rayleigh damping coefficients. The material parameters $E_m$, $G_m$ and $f_{cm}$ correlate positively with the PGA at failure, with the strongest correlation detected for $E_m$. The parameters $E_m$, $G_m$ and $f_{cm}$ were set as correlated when the LHS sample was prepared; it was therefore expected that all correlate in a similar manner with the PGA at failure. A positive correlation between the PGA at failure and drift limit values was expected for models that developed mainly in-plane failures. The shear drift at collapse had a more significant correlation with the PGA at failure than the flexural drift at collapse. Rayleigh damping ratio was expected to be correlated with the PGA at failure for any failure mode of non-linear dynamic analyses. Floor stiffness and floor-to-wall connection parameters did not correlate with the PGA at failure, even though the simulations led to an almost equal number of out-of-plane and in-plane collapses. This was because the out-of-plane failures were located in the walls spanning parallel and failing out-of-plane perpendicular to the direction in which the beams spanned, and as such were not influenced by these parameters.

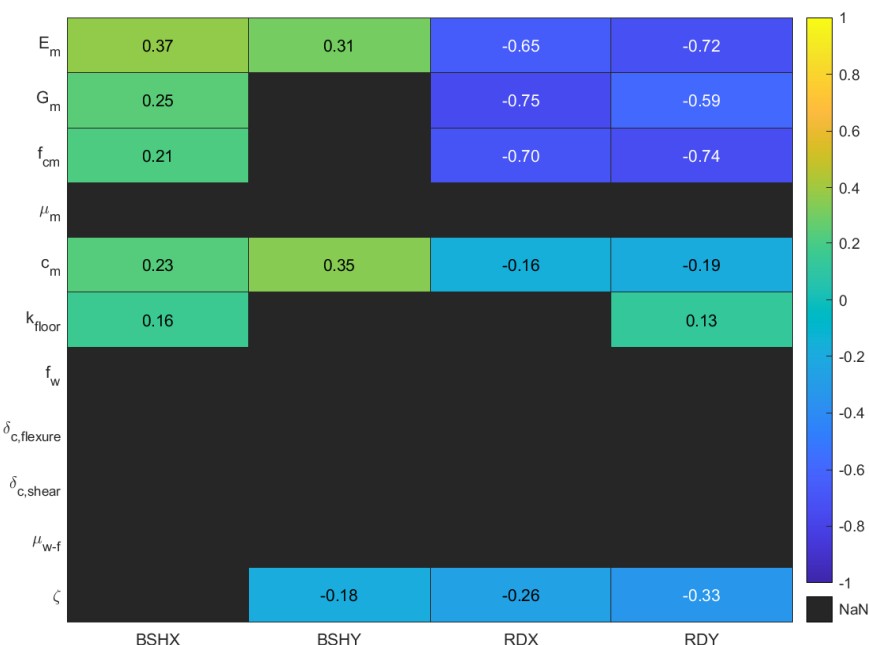

**Figure 9.** Holsteiner Hof model Spearman correlation matrix for PGA 0.21 g.

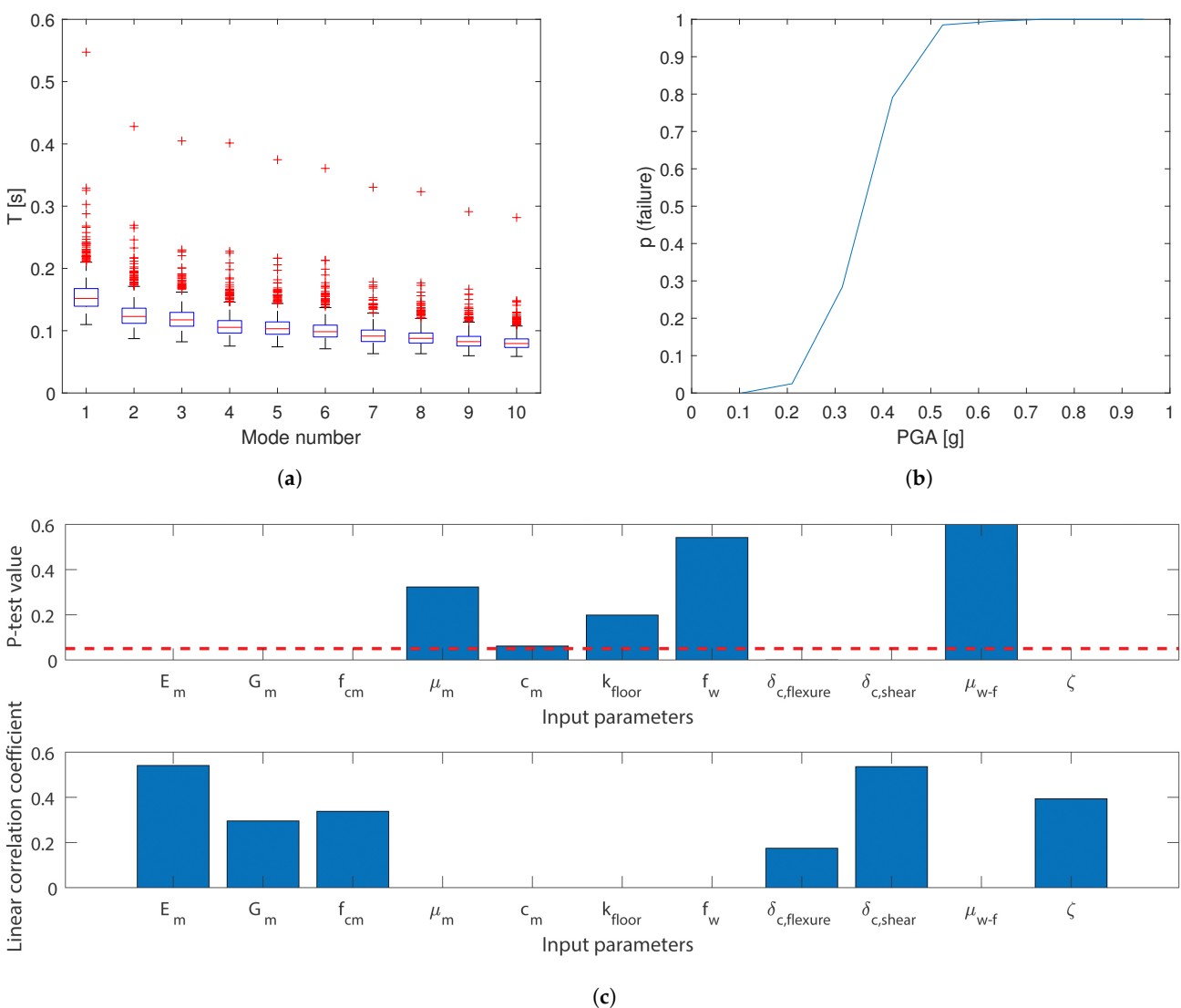

**Figure 10.** Holsteiner Hof model: (**a**) Distribution of modal periods. (**b**) Fragility curve. (**c**) Correlations between the PGA at failure and input parameters.

The failure PGA was not the only value impacted by the material and modelling parameters, as the portion of the building that fails was also influenced. The distribution of the types of failure can be seen in Figure 11. Interestingly, the number of the out-of-plane and in-plane failures was divided evenly. In-plane failures were again split between flexural and shear failures, with flexural failures accounting for a larger portion. A majority of the failures were localized in the second floor piers spanning in the y-direction for both out-of-plane and in-plane failures. Still, failure location was detected in all other parts as well.

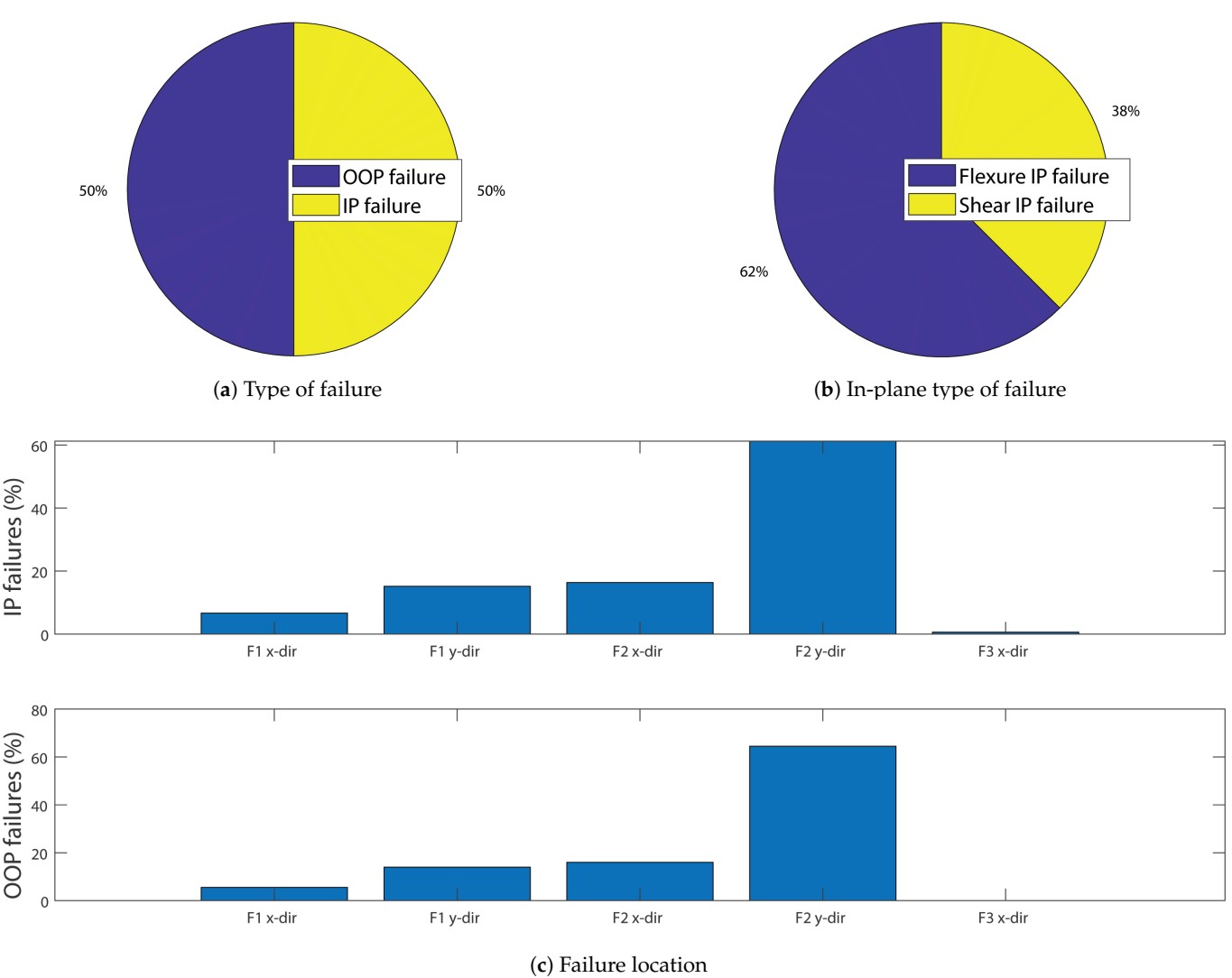

(**a**) Type of failure

(**b**) In-plane type of failure

(**c**) Failure location

**Figure 11.** Holsteiner Hof model failure statistics: (**a**) Type of failure. (**b**) In-plane type of failure. (**c**) Failure location.

### 4.2. Holsteiner Hof-Out-of-Plane Disabled and Rigid Connections

The Holsteiner Hof was again modelled as described in Section 2.1. The only difference was that the out-of-plane capability of the macro-element was disabled and connections were modelled as rigid. Therefore, both the seismic response parameters and the failure of the building were governed by the in-plane behaviour.

#### 4.2.1. Seismic Response Parameters

For each of the 400 sets of analyses, the IDA curves are plotted in Figure 12. The scatter of base shear and roof displacements was similar to the model with out-of-plane enabled.

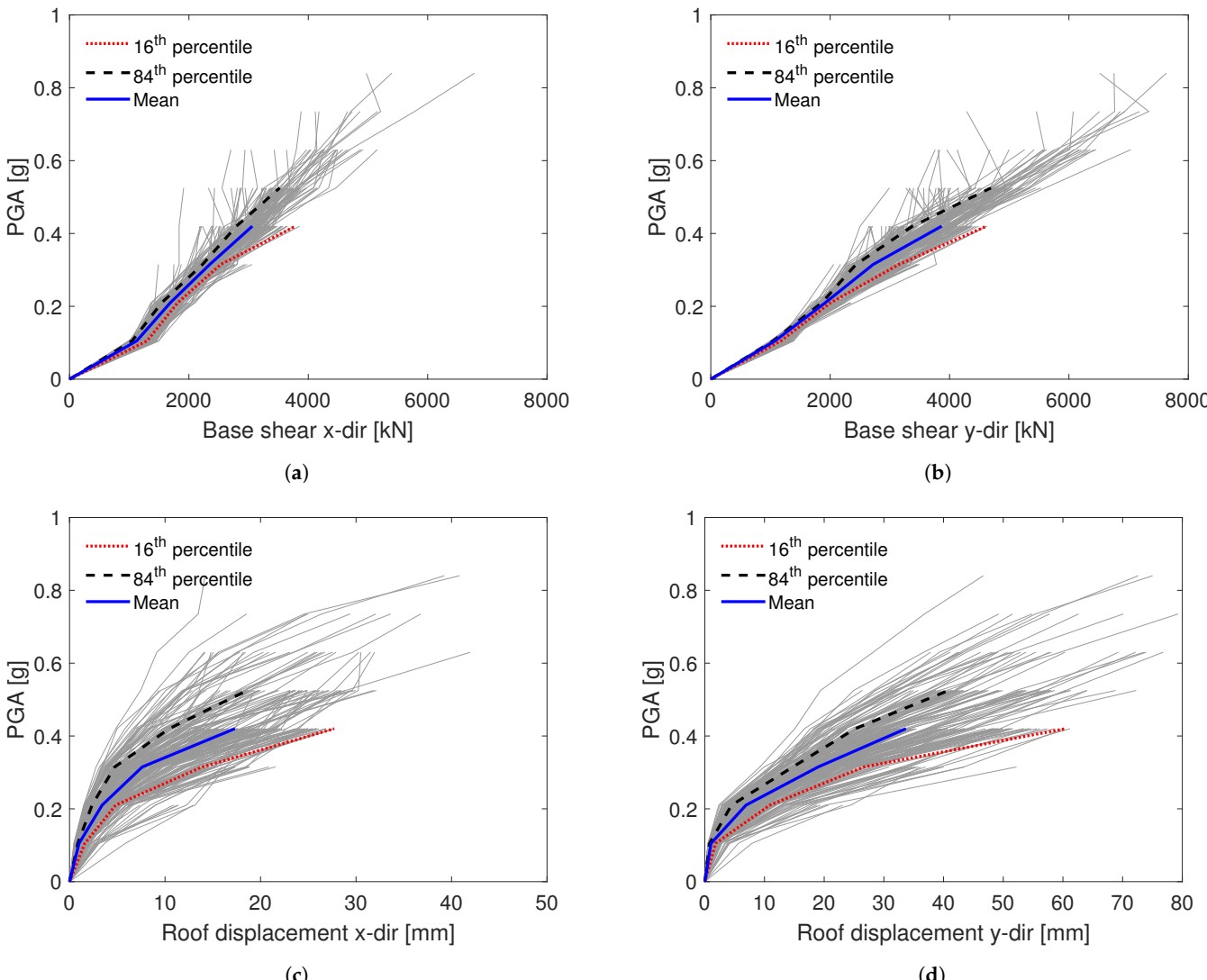

**Figure 12.** Holsteiner Hof model (with out-of-plane disabled and rigid connections) IDA curves displaying the maximum values of seismic demand parameters: (**a**) Total base shear in x-direction. (**b**) Total base shear in y-direction. (**c**) Average roof displacement in x-direction. (**d**) Average roof displacement in y-direction.

The correlation between the seismic demand parameters and the input parameters for the PGA of 0.21 g is displayed in Figure 13. For the selected PGA, the values were similar to the model with out-of-plane enabled, except that no influence of the floor stiffness was observed.

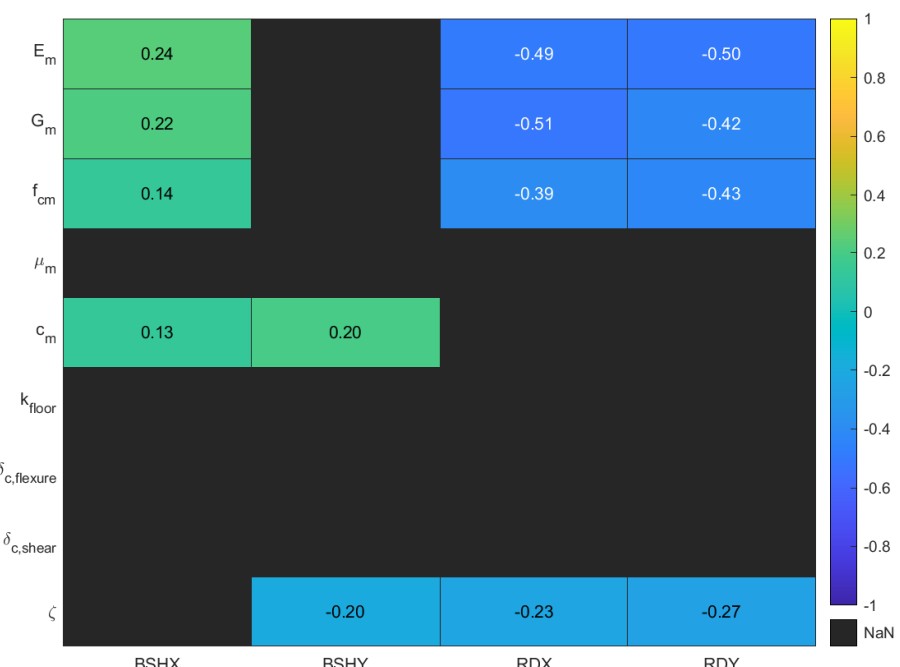

**Figure 13.** Holsteiner Hof model (with out-of-plane disabled and rigid connections) Spearman correlation matrix for PGA 0.21 g.

### 4.2.2. Failure Analysis

Figure 14 shows the results of 400 IDAs in terms of modal periods, fragility curve and the failure PGA correlations. Significant differences were observed at the fragility curved and discussed in the next section.

The difference with the model with out-of-planed enabled was observed in the portion of the building that fails. With out-of-plane disabled, in-plane was the only mode of the failure. As shown in Figure 15, in-plane failures were split between flexural and shear failures, with shear failures accounting for 64% of the failures. A majority of the failures were localized in the second floor piers spanning in y-direction, with significant number of failures as well in the first floor piers spanning in the same direction.

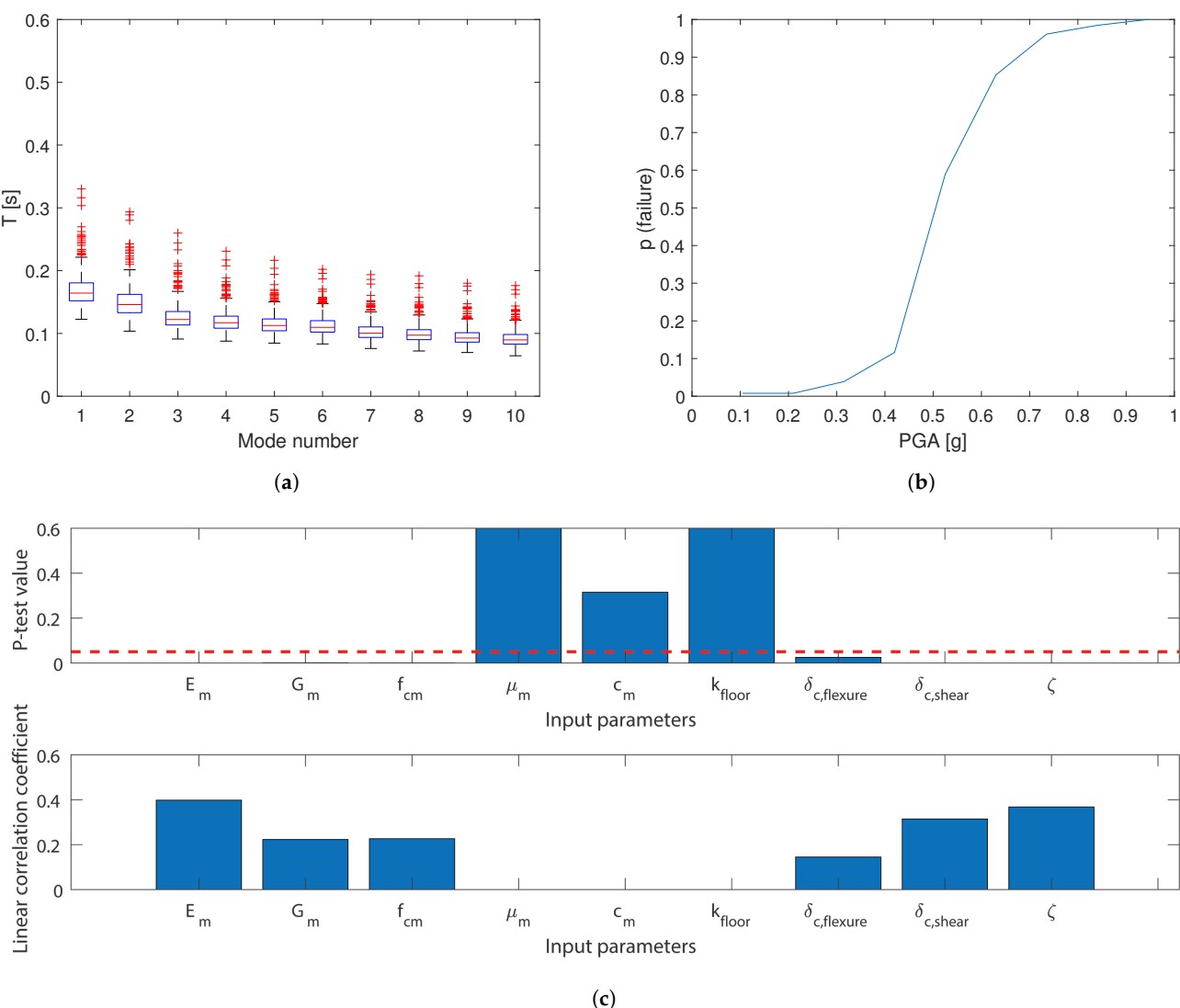

**Figure 14.** Holsteiner Hof model with out-of-plane disabled and rigid connections: (**a**) Distribution of modal periods. (**b**) Fragility curve. (**c**) Correlations between PGA at failure and input parameters.

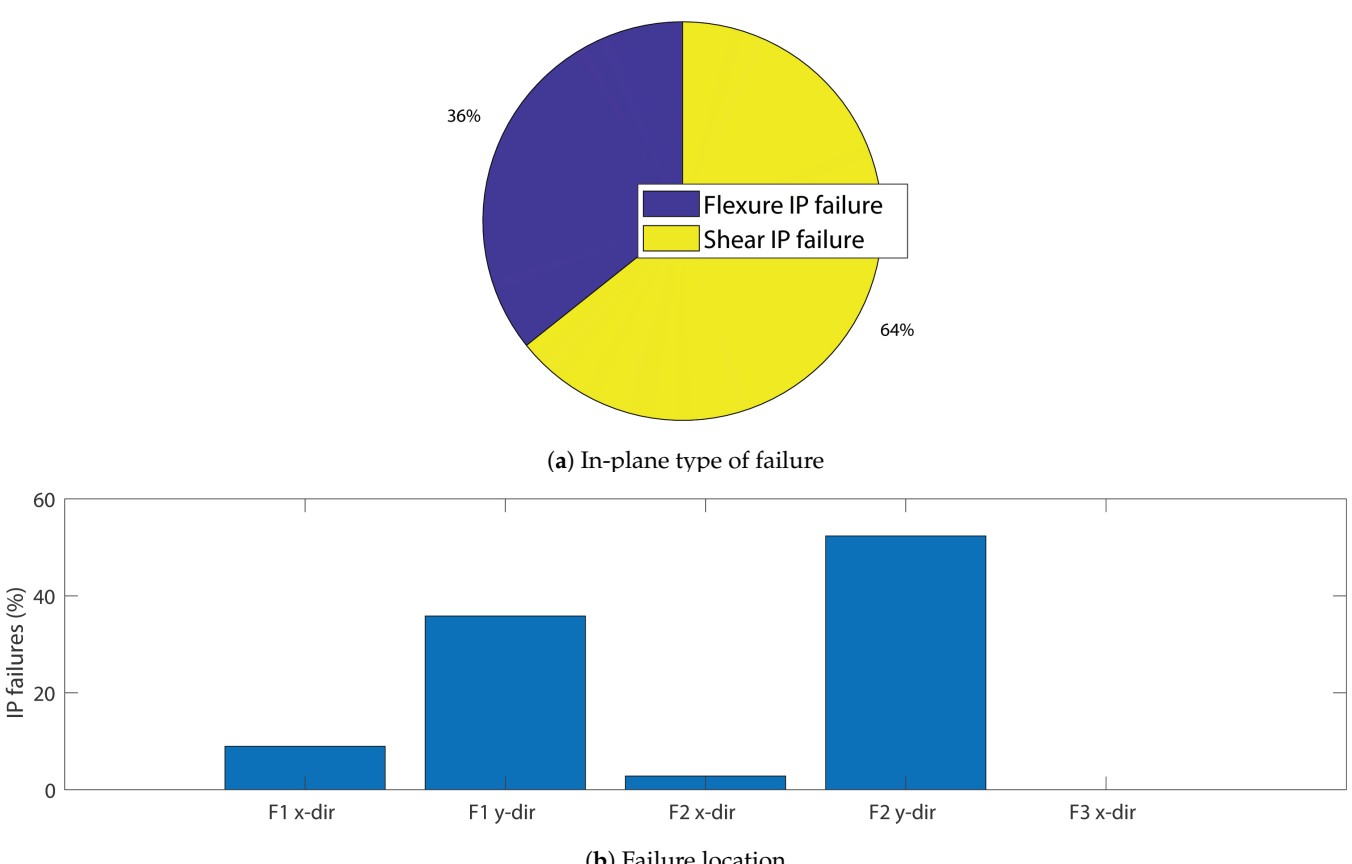

(**a**) In-plane type of failure

(**b**) Failure location

**Figure 15.** Holsteiner Hof model (with out-of-plane disabled and rigid connections) failure statistics: (**a**) In-plane type of failure. (**b**) Failure location.

### 4.3. Lausanne Malley

The Lausanne Malley building was modelled as described in Section 2.2. The walls were modelled using macroelements, the floors were modelled using orthotropic membrane elements, and the wall-to-wall and floor-to-wall connections were modelled using zero-length elements and non-linear material models.

### 4.3.1. Seismic Response Parameters

Figure 16 shows the IDA curves for each of the 400 sets of analyses. In this figure, the chosen output variable can be seen to evolve with the PGA increment, which is plotted together with the mean curve and 16th and 84th percentile curves. A similar scatter pattern was observed in both directions.

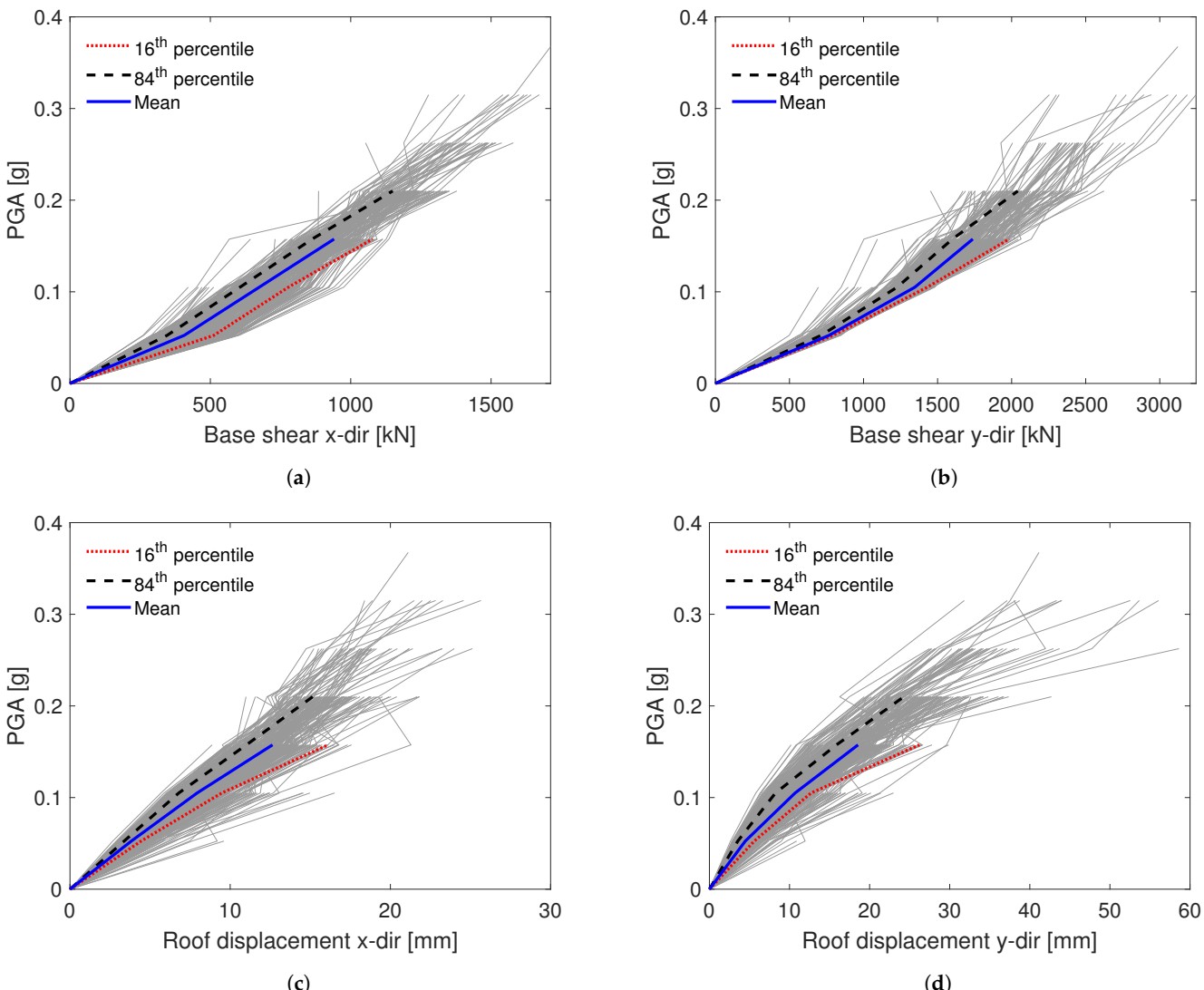

**Figure 16.** Lausanne Malley model IDA curves for maximum values of seismic demand parameters: (**a**) Total base shear in x-direction. (**b**) Total base shear in y-direction. (**c**) Average roof displacement in x-direction. (**d**) Average roof displacement in y-direction.

Figure 17 shows the correlation between the seismic demand parameters and the input parameters for the PGA of 0.105 g, chosen to provide sufficiently large seismic demand parameters without triggering too many failures. For clarity, only the Spearman rank correlation values for those parameters with a *p*-value less than 5% are displayed. Again, the strongest negative correlation was between the roof displacements and modulus of elasticity, shear modulus, and compressive strength, followed by Rayleigh critical damping ratio and cohesion. Modulus of elasticity, shear modulus, compressive strength, floor stiffness and cohesion showed positive correlations with base shear. Unlike in the case of Holsteiner Hof, there was a positive correlation between floor-to-wall friction coefficient and the base shear, and the negative correlation with the roof displacement in the y-direction. The reason for this lies in the fact that for the Lausanne Malley building, even for the low PGA values in some cases, floor-to-wall connection capacity is exceeded as out-of-plane behavior initiates. However, in most of the cases, the floor stiffness is still a more relevant value. Since the PGA was too low to trigger in-plane element failures, in-plane drifts at collapse did not correlate with the parameters.

### 4.3.2. Failure Analysis

Figure 18 shows the results of 400 IDAs for the Lausanne Malley model in terms of modal periods, fragility curve and the failure PGA correlations. At the start of each IDA, modal periods were determined to calculate the Rayleigh damping coefficients and evaluate the stiffness of the building. The material parameters $E_m$, $G_m$ and $f_{cm}$ correlated positively with the PGA at failure, with the strongest correlation detected for $E_m$. It is important to note that these parameters were set as correlated when the LHS sample was prepared. A correlation between the limit drift values was expected for the in-plane failures. Since the Lausanne Malley building was more prone to failing in flexure, flexural drift at collapse more significantly influenced the PGA at failure than the shear drift at collapse. It was also expected that the Rayleigh damping ratio and the failure PGA will be correlated at failure for any failure mode for non-linear dynamic analyses. This is especially true for buildings that tend to fail out-of-plane, because the damping ratio is known to influence the out-of-plane displacement significantly [16]. Here, the floor stiffness correlated with the PGA at failure. However, the floor-to-wall friction coefficient was more significantly correlated with the PGA at failure, as was expected in a building with such a significant percentage of out-of-plane failures in both directions.

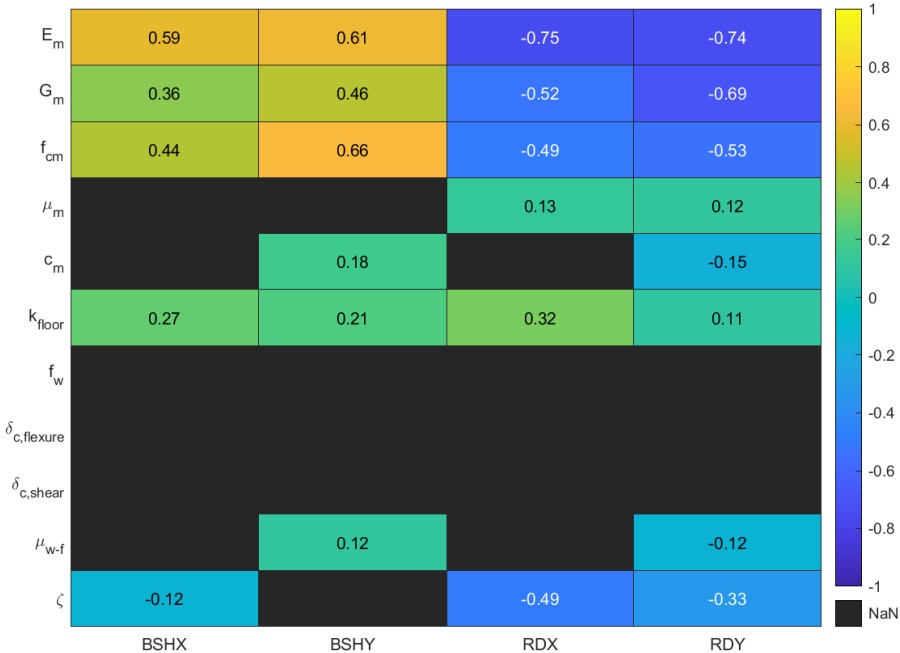

**Figure 17.** Lausanne Malley model Spearman correlation matrix for PGA 0.105 g.

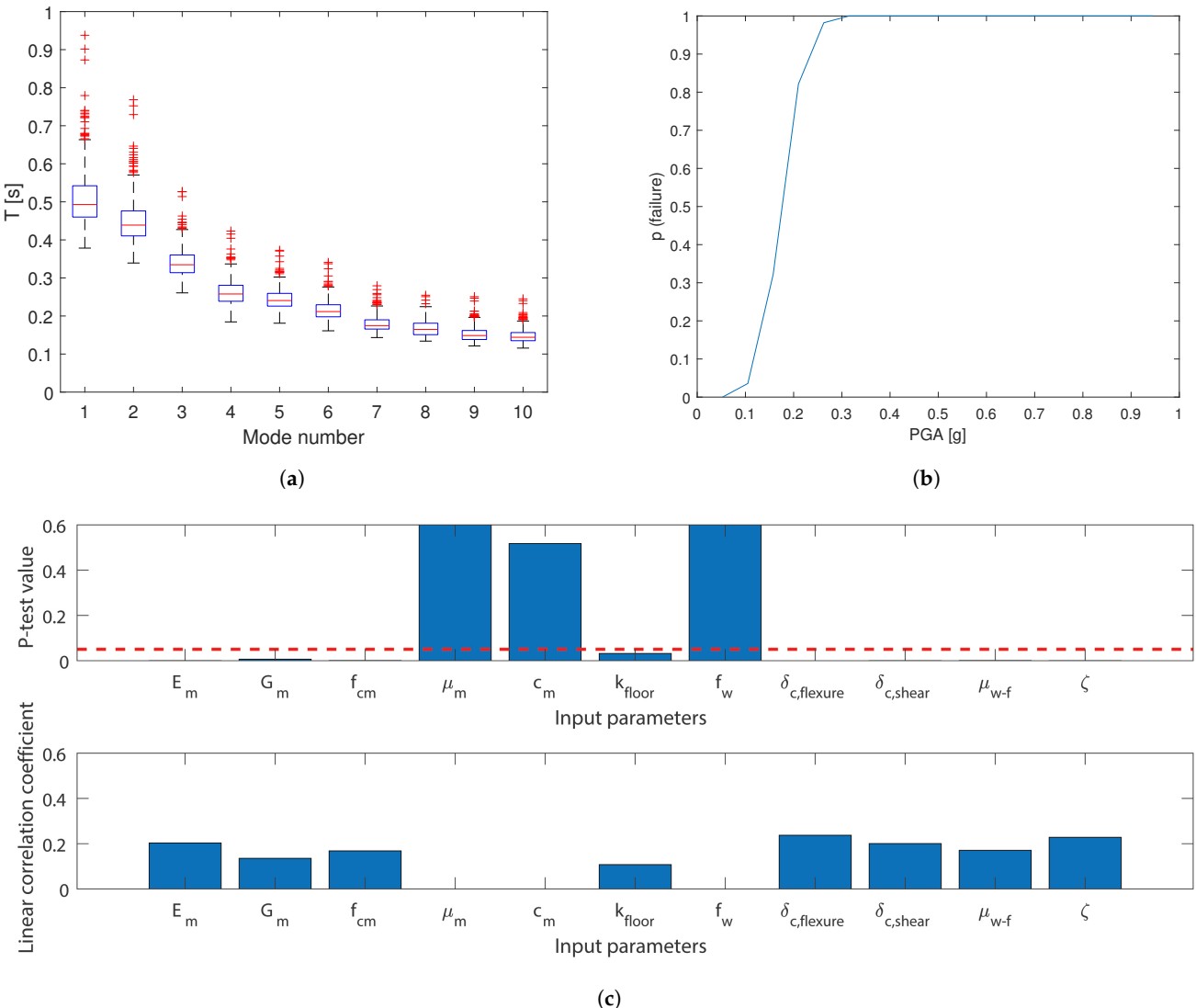

**Figure 18.** Lausanne Malley model: (**a**) Distribution of modal periods. (**b**) Fragility curve. (**c**) Correlations between PGA at failure and input parameters.

The part of the building that fails was also influenced by selected material and modelling parameters. The distribution of the types of failure can be seen in Figure 19. Out-of-plane is the dominant failure mode, accounting for 92% of the total failures. In-plane failures were again dominated by flexural failures, accounting for 97% of total in-plane failures. This was expected for a tall and slender building, especially for upper floors with rather low axial loads on the piers. A majority of the failures, either in-plane or out-of-plane, were localized in the fifth floor piers. Twice as many in-plane failures occurred in the piers in x-direction than in the y-direction, while this difference was smaller for the out-of-plane failure modes. Regardless, in all the other storeys, the failure location was occasionally detected in both directions. This scatter of failures emphasizes the sensitivity of numerical simulations.

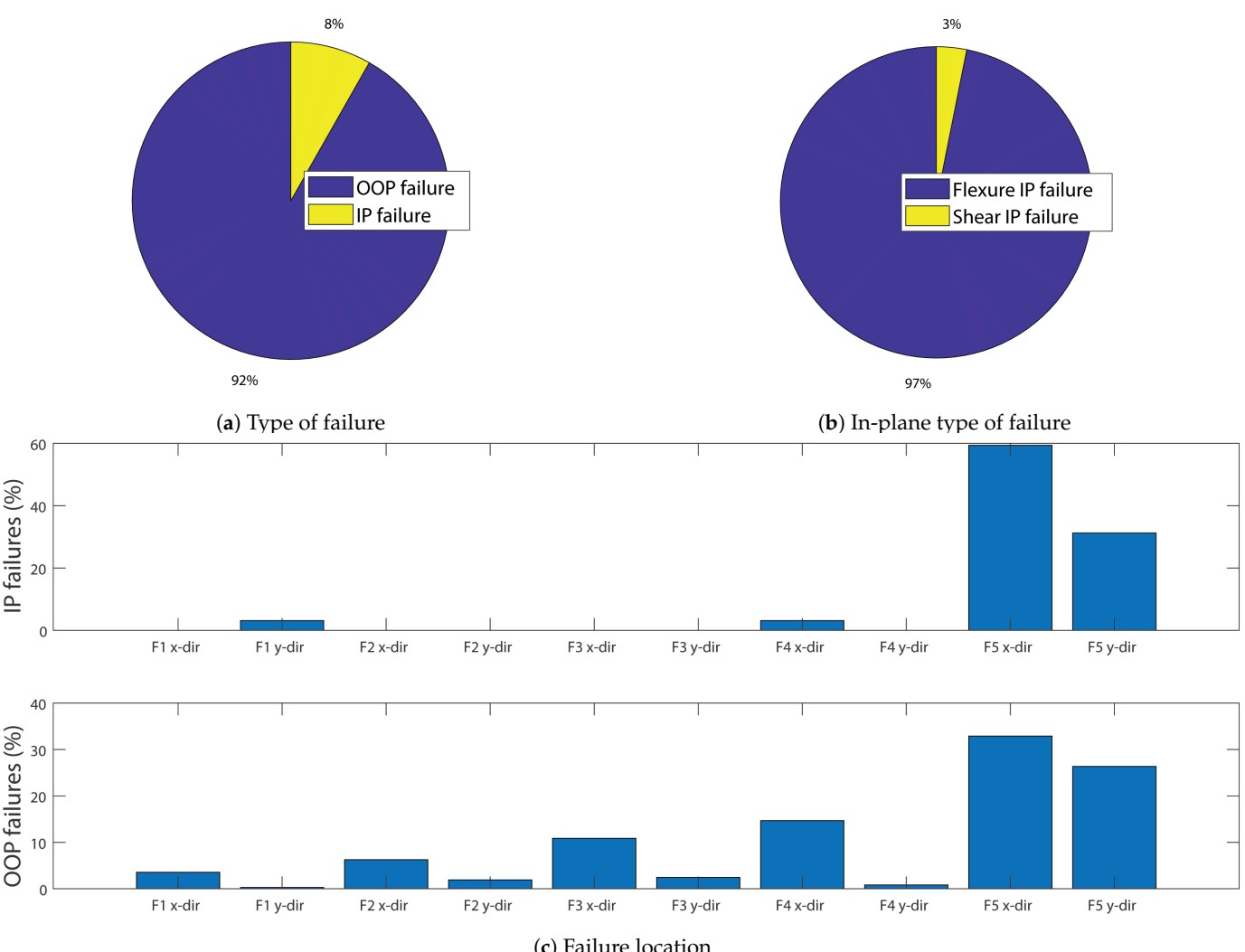

**(a)** Type of failure          **(b)** In-plane type of failure

**(c)** Failure location

**Figure 19.** Lausanne Malley model failure statistics: (**a**) Type of failure. (**b**) In-plane type of failure. (**c**) Failure location.

### 4.4. Lausanne Malley-Out-of-Plane Disabled and Rigid Connections

Lausanne Malley was again modelled as described in Section 2.2. The only difference was that the out-of-plane capability of the macro-element was disabled and connections were modelled as rigid. Therefore, both the seismic response parameters and the failure of the building were governed by the in-plane behaviour.

#### 4.4.1. Seismic Response Parameters

For each of the 400 sets of analyses, the IDA curves are plotted in Figure 20. The scatter of base shear and roof displacements was similar to the model with out-of-plane enabled.

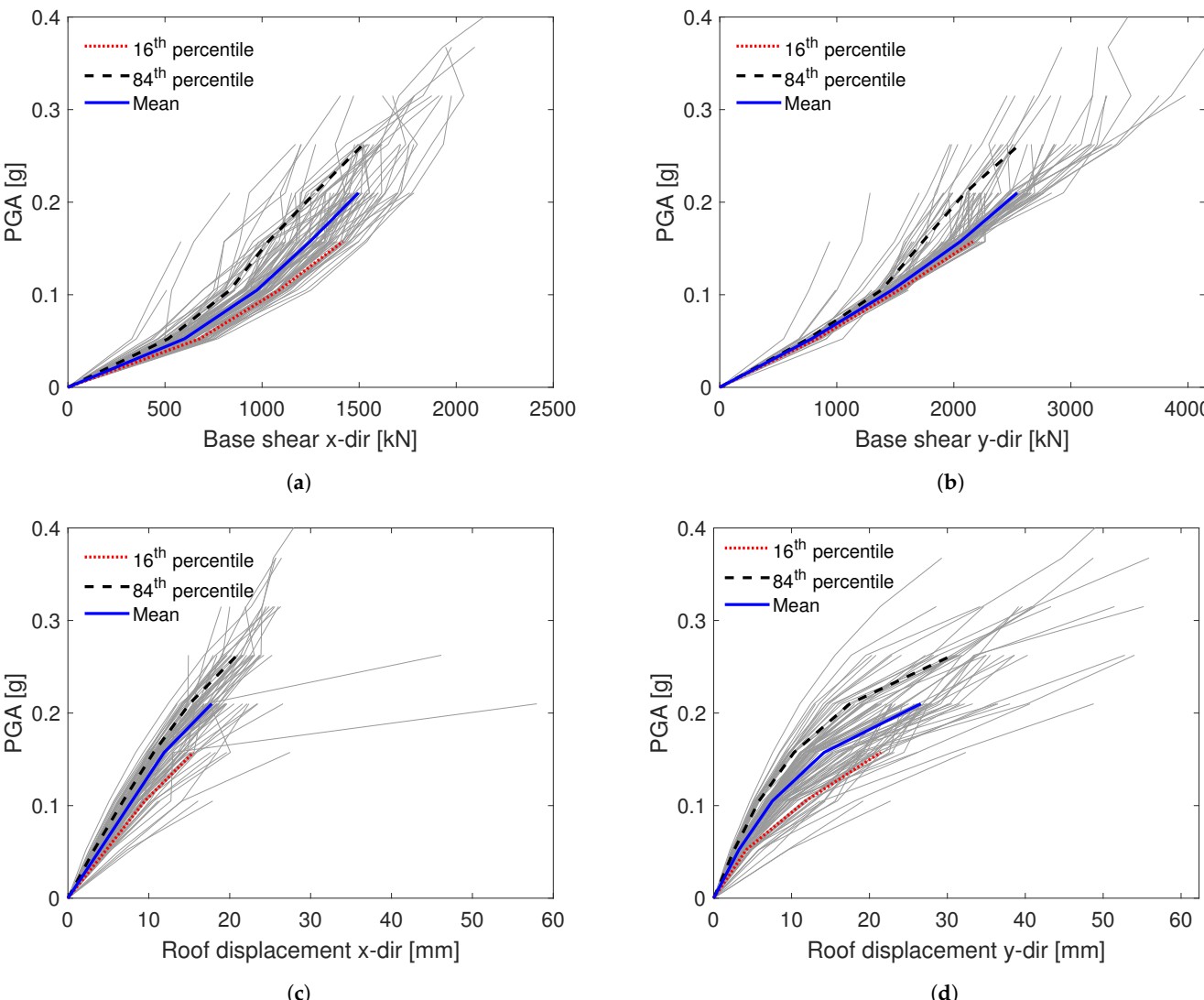

**Figure 20.** Lausanne Malley model (with out-of-plane disabled and rigid connections) IDA curves displaying the maximum values of seismic demand parameters: (**a**) Total base shear in x-direction. (**b**) Total base shear in y-direction. (**c**) Average roof displacement in x-direction. (**d**) Average roof displacement in y-direction.

The correlation between the seismic demand parameters and the input parameters for the PGA of 0.105 g is displayed in Figure 21. For the selected PGA, the values are very similar to the model with out-of-plane and non-linear connections.

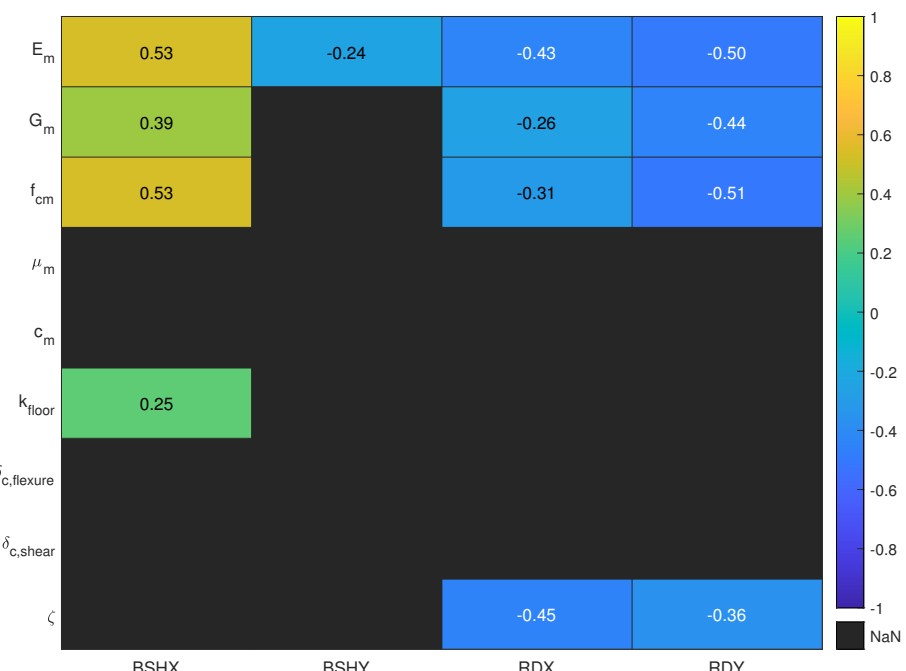

**Figure 21.** Lausanne Malley model (with out-of-plane disabled and rigid connections) Spearman correlation matrix for PGA 0.105 g.

### 4.4.2. Failure Analysis

Figure 22 shows the results of 400 IDAs in terms of modal periods, fragility curve and the failure PGA correlations. Significant differences to the model with out-of-plane enabled and non-linear connections were detected. First, in terms of the seismic fragility, and then in the parameters correlated with the PGA at failure. Floor-to-wall friction connection was replaced with the rigid connection, and the floor stiffness was no longer correlated with the PGA at failure. Remaining correlated parameters were the same as in the model with out-of-plane enabled, but with higher correlation coefficients. This was explained by the fact that once the out-of-plane failure mechanisms were disabled, the in-plane mechanisms with which they were associated became predominant.

The distribution of the types of failure can be seen in Figure 23. In-plane failures were again dominated by flexural failures, accounting for 100% compared to 92% in the model with the out-of-plane enabled and non-linear connections. The scatter decreased, focusing most of the failures in the y-direction of the fifth storey.

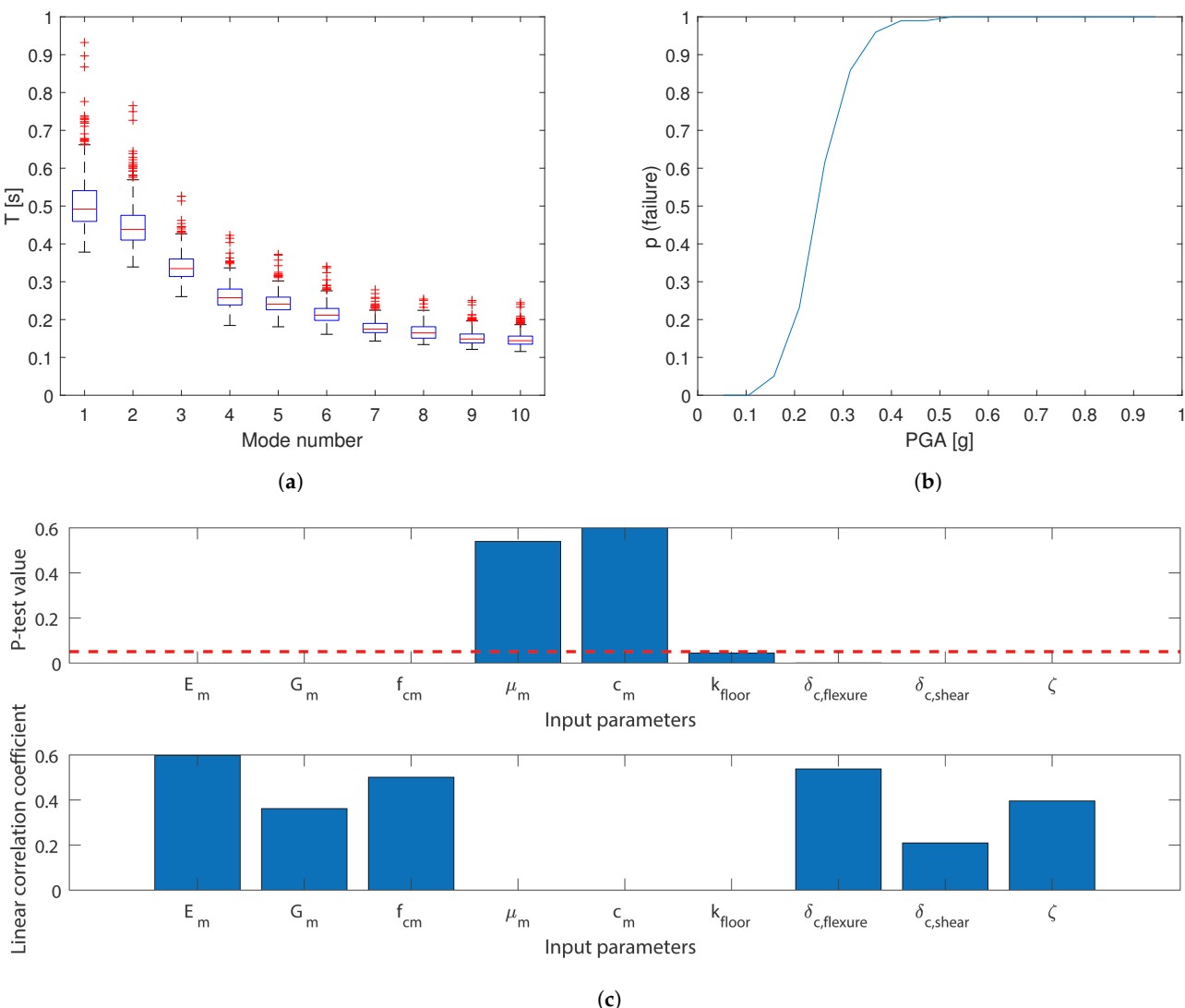

**Figure 22.** Lausanne Malley model with out-of-plane disabled and rigid connections: (**a**) Distribution of modal periods. (**b**) Fragility curve. (**c**) Correlations between PGA at failure and input parameters.

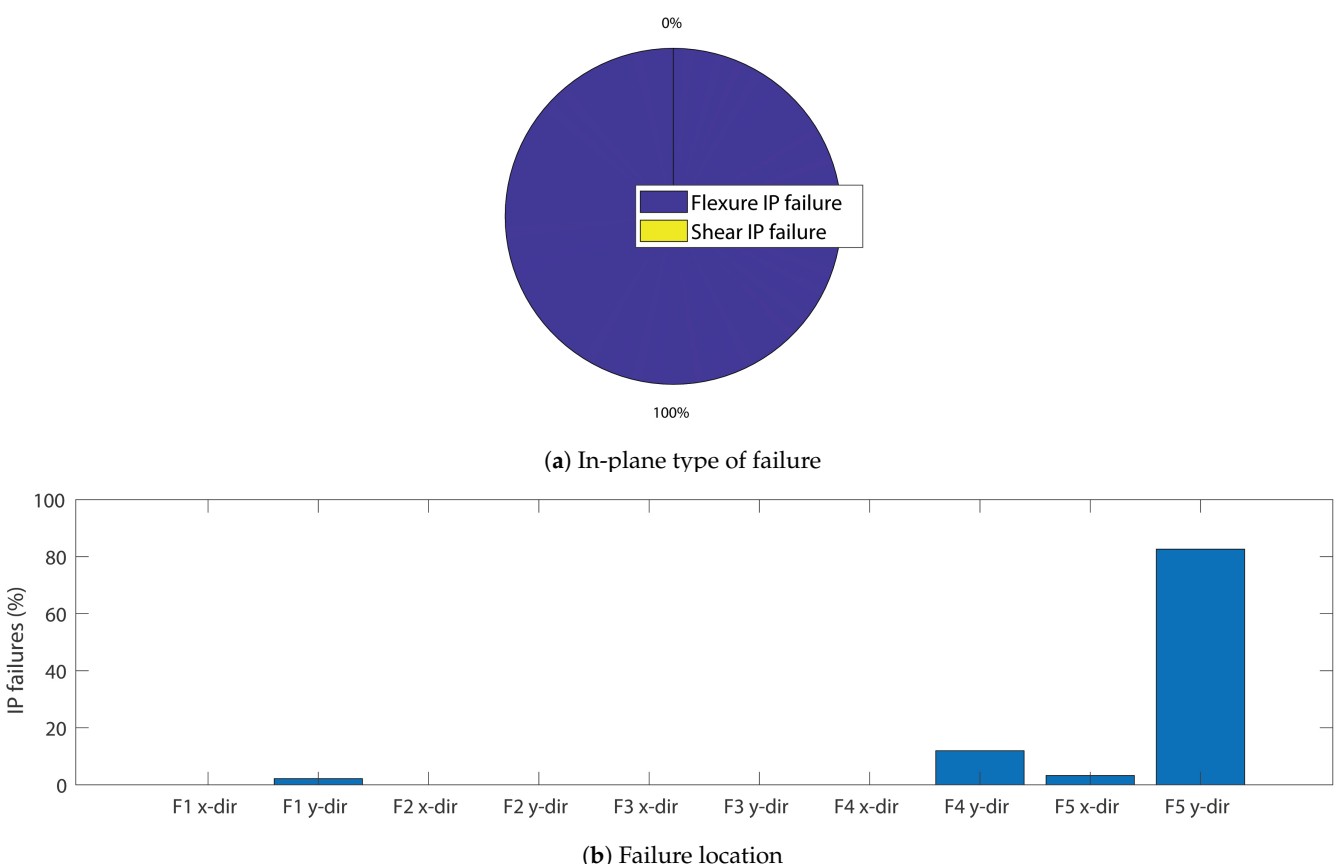

(**a**) In-plane type of failure

(**b**) Failure location

**Figure 23.** Lausanne Malley model (with out-of-plane disabled and rigid connections) failure statistics: (**a**) In-plane type of failure. (**b**) Failure location.

## 5. Discussion

This work highlights the importance of considering all sources of material and modelling uncertainty since rather small differences in the material and modelling parameters led to rather different results. Firstly, differences here could produce a different PGA at failure, which was already visible from the fragility curves. Secondly, they could produce different modes and locations of failure; the difficulty in predicting the correct mode of failure was especially evident for the Holsteiner Hof building, where the probability of out-of-plane and in-plane failures were evenly divided. Furthermore, the in-plane failure modes comprised both shear and flexural failures, with 62% of the in-plane failure modes being flexural, and both the flexural and shear drifts at collapse showing a clear correlation with PGA at failure, together with the Rayleigh critical damping ratio, modulus of elasticity, shear modulus and compressive strength.

Although the Holsteiner Hof had a high number of out-of-plane failures, there was no correlation detected between the probability of out-of-plane failures and floor stiffness or floor-to-wall connection effectiveness. This is because the floors of the building span in the y-direction, whereas Figure 11 showed that the majority of out-of-plane failures were located in the second storey walls in the y-direction. Since these walls fail out-of-plane in the x-direction and the timber floor beam span in the opposite direction, the floor-to-wall connection does not influence the out-of-plane failure unless retrofit interventions provide also floor-to-wall connections in this direction.

To link our work with previous studies where EFM were used to compute fragility functions without accounting for out-of-plane failures, the stochastic analyses were repeated with the out-of-plane capability of the macroelement disabled and floor-to-wall and wall-to-wall connections modelled as rigid. The Holsteiner models 84th percentile IDA

curves are compared in Figure 24. The results in terms of the base shear are almost identical, while the difference in the roof displacements progressed with the PGA increment.

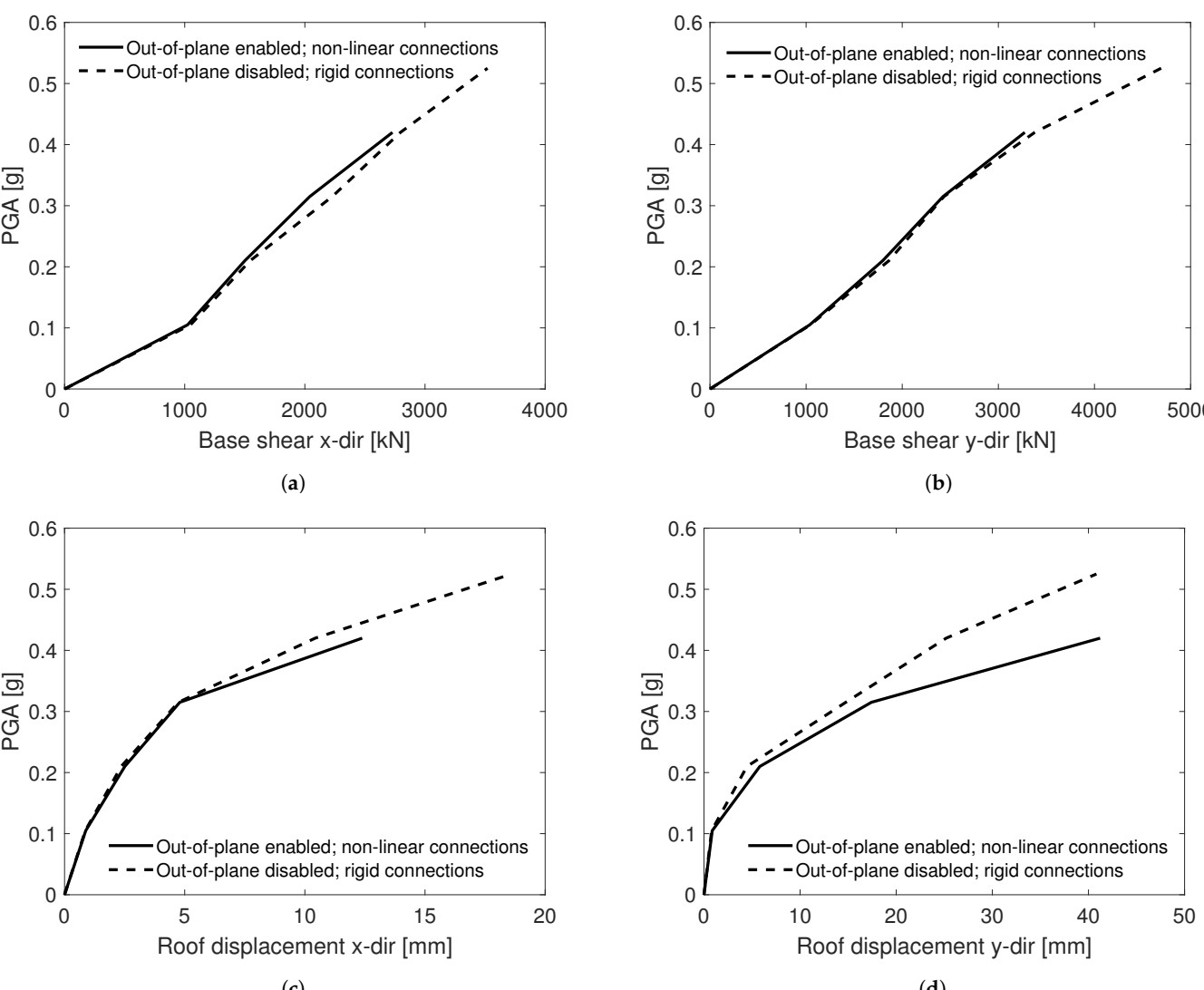

**Figure 24.** Holsteiner Hof comparison of IDA curves for models with: (i) out-of-plane enabled and non-linear connections, and (ii) out-of-plane disabled and rigid connections, displaying the maximum values of seismic demand parameters: (**a**) Total base shear in x-direction. (**b**) Total base shear in y-direction. (**c**) Average roof displacement in x-direction. (**d**) Average roof displacement in y-direction.

Comparison of fragility curves showed that the model with out-of-plane behaviour and non-linear connections was significantly more fragile than the model without out-of-plane behaviour and with rigid connections, having higher probability of failure for each PGA level as shown in Figure 25. The difference was emphasized for lower PGA levels, where a significant occurrence of the early out-of-plane failure was observed. The difference in failure type and location was significantly influenced by the ability of the macroelement to simulate the out-of-plane behaviour and presence of non-linear connections. Whereas the first model failed in flexure in 62% of the cases, the second failed in shear in 64% of the cases. Although failure was still most common to occur in the second storey, having disabled the out-of-plane behaviour and non-linear connections, the number of failures located in the first storey increased.

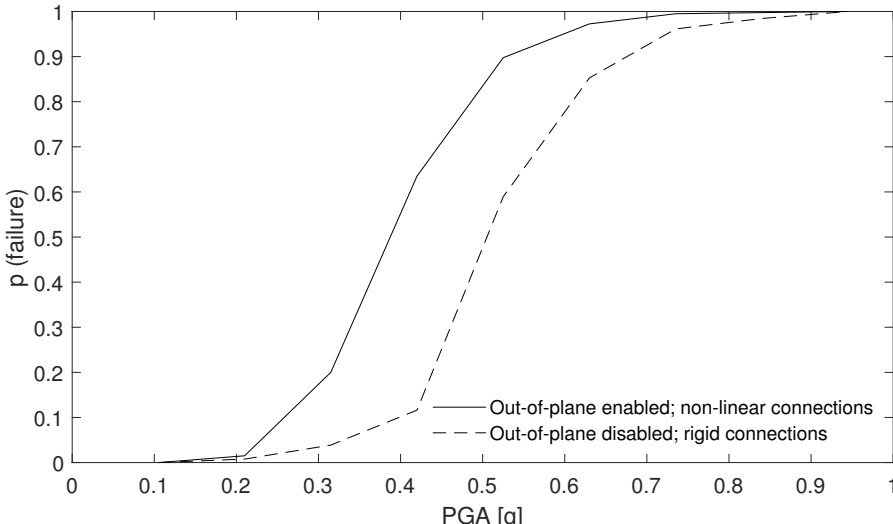

**Figure 25.** Comparison of Holsteiner Hof fragility curves for models with: (i) out-of-plane enabled and non-linear connections, and (ii) out-of-plane disabled and rigid connections.

A different effect of modelling uncertainties was observed in the Lausanne Malley building, which is tall and slender. As such, it failed out-of-plane 92% of the time. A dominant out-of-plane behaviour is shown in Figure 19, wherein failures were distributed in both directions and scattered through floors, but nevertheless were mostly concentrated in the fifth storey. This scatter throughout the floors indicates uncertainty on its own, and since out-of-plane was such a significant mode of failure in both directions, this also highlights the importance of floor stiffness and floor-to-wall connections, in particular. This can be seen in Figure 18, where a correlation with floor stiffness could be observed along with a stronger correlation with floor-to-wall friction. This result was expected, since only when the floor-to-wall connection capacity was not exceeded, the full stiffness of the floor could have been exploited. This correlation was on the same order of importance as damping ratio, flexure drift at collapse and modulus of elasticity—all values that were expected to significantly correlate with the PGA at failure.

Lausanne Malley models with and without out-of-plane behaviour and non-linear connections also showed differences in performance in terms of the 84th percentile IDA curves, as shown in Figure 26. The differences with the PGA increment were the most significant in the roof displacements in y-direction.

Comparison of the fragility curves for the Malley Lausanne building showed that the model with out-of-plane behaviour and non-linear connections was significantly more fragile, having higher probability of failure for each PGA level, but especially for the lower PGA as shown in Figure 27. The difference was pronounced for lower PGA levels due to a significant occurrence of early out-of-plane failure, whereas the model with only in-plane behaviour started experiencing collapses only for higher PGA levels, when the in-plane capacity was exceeded. Having out-of-plane behaviour and non-linear connections disabled, the floor-to-wall friction coefficient was no longer used and the floor stiffness did not show a correlation with the PGA at failure. This was different in the case of the model with out-of-plane and non-linear connections, where they were correlated with the PGA at failure, stronger for the case of the floor-to-wall friction. The other parameters that passed the P-test remained the same, but with their correlation coefficients increased as a result of governing failure mechanism becoming predominant. Whereas the model with out-of-plane enabled and non-linear connections failed in flexure in 92% of the cases, the model with out-of-plane disabled and rigid connections failed in flexure in 100% of the cases. However, differences were detected in the failure location. In the model with out-of-plane disabled and rigid connections, failure was less scattered and occurred in the fifth storey in most of the cases.

All analyses were carried out with only one ground motion record (see Section 2.5). Some of the findings presented in this study might be record dependent. However, due to the rather broad-band frequency content of the Montenegro Albatros 1979 record, it is expected that the qualitative conclusions are applicable to a wide range of records. Dolsek found that both record-to-record variability (aleatory variability) and epistemic uncertainty influence the IDA curves [7]. While the aleatory variability influenced the dispersion in the IDA curves, the epistemic uncertainty affected primarily the collapse capacity [7]. Our work focused on the epistemic uncertainties, but in future work, the aleatory and the epistemic uncertainties could be combined using the same methodology to confirm the validity of using a record with a rather broad-band frequency content.

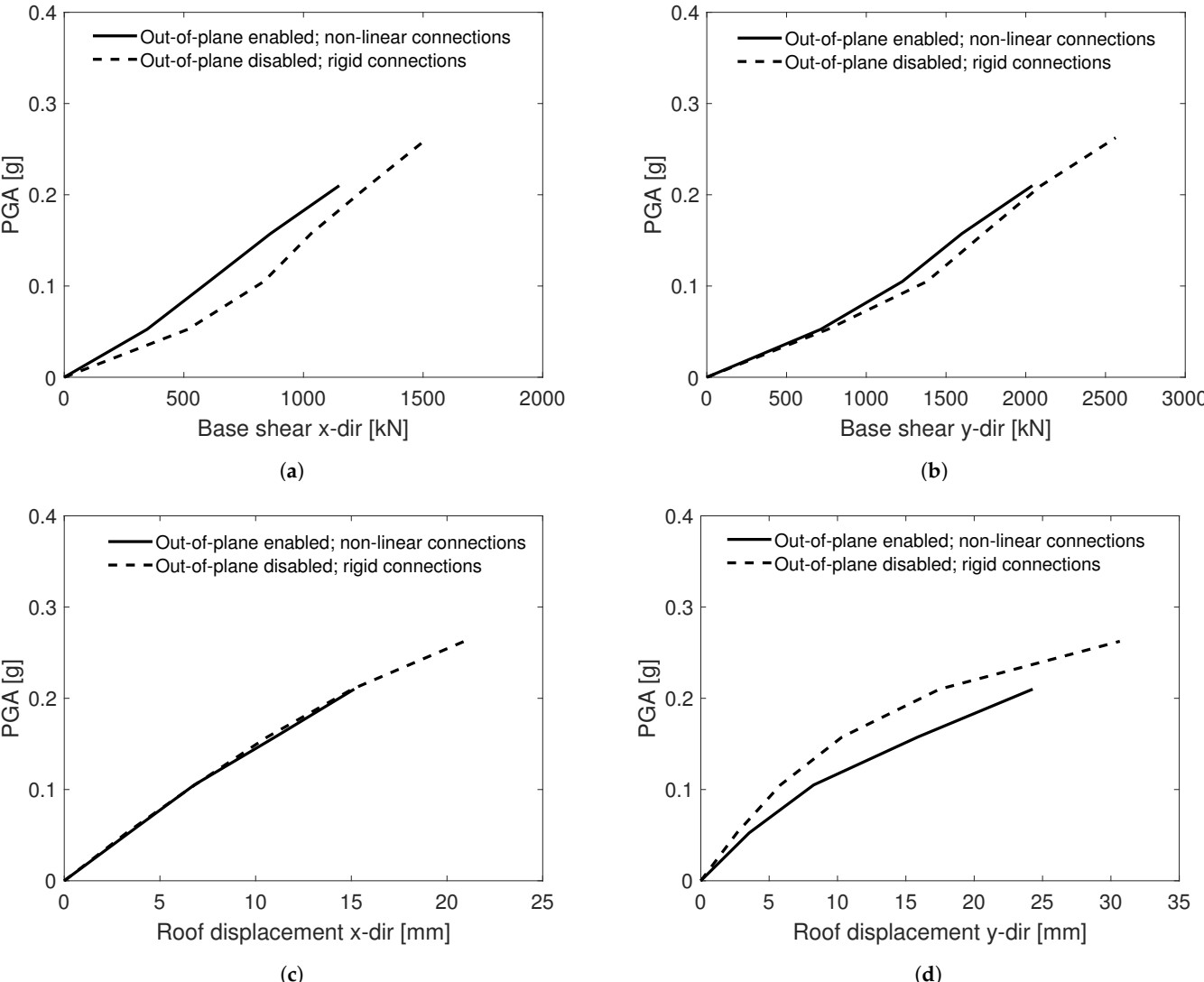

**Figure 26.** Lausanne Malley comparison of IDA curves models with: (i) out-of-plane enabled and non-linear connections, and (ii) out-of-plane disabled and rigid connections, displaying the maximum values of seismic demand parameters: (**a**) Total base shear in x-direction. (**b**) Total base shear in y-direction. (**c**) Average roof displacement in x-direction. (**d**) Average roof displacement in y-direction.

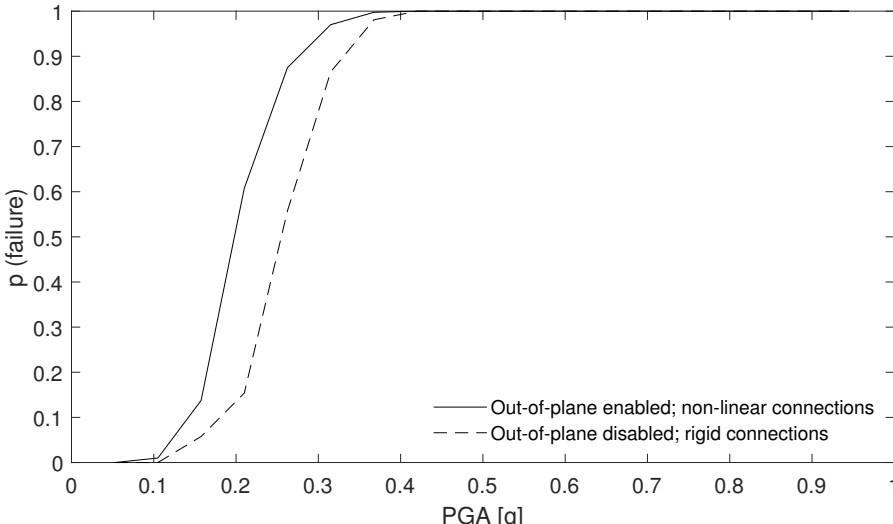

**Figure 27.** Comparison of Lausanne Malley fragility curves for models with: (i) out-of-plane enabled and non-linear connections, and (ii) out-of-plane disabled and rigid connections.

## 6. Conclusions

Scientists and practitioners alike often model unreinforced masonry buildings using the equivalent frame model (EFM) approach. This method is affected by numerous epistemic uncertainties, commonly handled by adopting conservative deterministic values. However, any deterministic approach bears the risk that possible damage and collapse mechanisms are overlooked. Therefore, we proposed a methodology to account for epistemic uncertainties and applied it to two masonry buildings representative of their categories: a stiff monumental historical masonry building, and a tall and slender residential masonry building. Each building was modelled twice —first with the out-of-plane capability of the macroelement used for modelling each wall enabled and with non-linear wall-to-wall and floor-to-wall connections, and then with the out-of-plane capability disabled and rigid connections. We used Latin Hypercube Sampling to create 400 sets of 11 material and modelling parameters, distributed according to normal and lognormal distributions, to account for epistemic uncertainties. Then for each model and each set of parameters, we carried out an IDA in OpenSEES using a newly developed macroelement that can couple in-plane and out-of-plane responses until the building failed or collapsed. Failure mode and location were analysed according to a set of criteria considering the loss of equilibrium due to out-of-plane failure of some element(s) or the progressive loss of lateral force capacity due to in-plane failure of walls. The response of each modelling approach was evaluated in terms of the seismic response parameters—base shear and roof displacement, correlations of seismic response parameters with input parameters, seismic fragility curves, correlations of failure PGA with input parameters, and statistical distributions of failure modes and locations.

The factors that clearly impact the failure PGA for each building typology were the material parameters (modulus of elasticity, shear modulus, compressive strength) and the EFM modelling parameters (limit drift values, Rayleigh damping ratio). The floor stiffness and floor-to-wall friction coefficient also impacted the failure PGA if the model developed a dominant out-of-plane failure mode. The floor stiffness was second in importance to the floor-to-wall friction coefficient. A similar effect was observed by Vanin et al. [16]. The importance of the floor stiffness is lower than the force capacity of the floor-to-wall connection if the latter is exceeded because the stiffness of the floor cannot be exploited. On the other hand, for PGA levels farther from failure, the floor-to-wall connection force capacity is not exceeded and the seismic response parameters correlate with the floor stiffness. Hence, the floor stiffness has only a large influence on the results, if the connection between floors and walls is effective enough to fully exploit the contribution of floors. In unstrengthened building configurations this force capacity is typically governed by friction

forces between floor beams and wall and is therefore rather small. While this finding is intuitive, it is often overlooked because the floor-to-wall connection is modelled as perfectly rigid. The effect of the floor-to-wall friction coefficient highlights therefore the importance of modelling the nonlinearity of the connections in historical masonry explicitly.

For both the Holsteiner Hof and Lausanne Malley buildings, disabling out-of-plane behaviour and making connections rigid produced significant differences in the seismic fragility, failure modes and locations. The difference in the seismic fragility was even pronounced for lower PGA levels, which for certain sets of material and modelling parameters, could already produce significant out-of-plane behaviour. The models with out-of-plane and non-linear connections disabled, started experiencing failures only for higher PGA levels, when the in-plane capacity would be reached. Therefore, we deduce that modelling the unreinforced masonry buildings with wooden floors without correctly modelling the out-of-plane behaviour leads to incorrect predictions of the seismic fragility, as well as on the mode and the location of the failure. This becomes especially relevant when the analysis is performed to design retrofitting interventions to prevent the detected failure modes.

As a result, we conclude that after the out-of-plane behaviour and non-linear connections are correctly modelled, the appropriate choice of material and modelling parameters, including those of non-linear connections, can impact the seismic demand parameters such as base shear and roof displacements, and the PGA at failure. Failure mode and location are, however, more sensitive to the material and modelling parameters than the seismic demand parameters. Our results highlight the importance of firstly correctly modelling out-of-plane behaviour and non-linear connections, and then performing a probabilistic analysis to account for uncertainties. Although choosing conservative deterministic parameters aims at a conservative approximation of the PGA at failure, there is a risk of overlooking damage mechanisms and their location. This can again be especially problematic if the goal of the analysis is to retrofit the building to prevent certain mechanisms. Further studies should focus on gaining deeper insights into accurately modelling non-linear connections for correctly simulating the global behaviour. Special attention should be paid to extend the modelling of the connections to account for the interaction between units of unreinforced masonry aggregates.

**Supplementary Materials:** The OpenSEES models used for producing the results presented in this paper as well as the sets of material and modelling parameters used for the IDAs are shared openly through the repository https://doi.org/10.5281/zenodo.4549230.

**Author Contributions:** Conceptualization, K.B. and I.T.; methodology, I.T.; software, F.V.; validation, K.B., F.V. and I.T.; formal analysis, I.T.; investigation, I.T.; resources, K.B. and F.V.; data curation, I.T.; writing—original draft preparation, I.T.; writing—review and editing, K.B. and F.V.; visualization, I.T.; supervision, K.B.; project administration, K.B.; funding acquisition, K.B. All authors have read and agreed to the published version of the manuscript.

**Funding:** The project was supported by the Swiss National Science Foundation through grant 200021_175903/1 Equivalent frame models for the in-plane and out-of-plane response of unreinforced masonry buildings.

**Institutional Review Board Statement:** Not applicable.

**Informed Consent Statement:** Not applicable.

**Acknowledgments:** We would like to express our gratitude to Dr. Pierino Lestuzzi for providing the data on the Lausanne Malley building.

**Conflicts of Interest:** The authors declare no conflict of interest.

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
