# Peer review of "Uncertainties in the Seismic Assessment of Historical Masonry Buildings"

_applsci, doi:10.3390/app11052280_

Round 1
Reviewer 1 Report
The present paper aims at investigate the role of mechanical parameters in the seismic assessment of masonry buildings. Authors performed an extensive numerical campaign on two cases studies, accounting for two modelling approaches and by considering the role of epistemic uncertainty in the nonlinear dynamic analyses and seismic fragility. A full set of results is provided. The paper is well written, well motivated and it can be accepted for publication. Few comments are provided in the attached PDF file, which can be considered by authors.

Reviewer 2 Report
The paper highlights the importance of considering all sources of material and modelling uncertainty in order to identify different modes and locations of failure when using the equivalent frame approach to assess the historical masonry building. The authors propose a methodology to account for epistemic uncertainties through the application to case studies of two masonry buildings representative of a stiff monumental building and a tall and slender residential building.
The key parameters influencing the performance of the buildings at failure, type of failure and failure location have been identified. In particular, the impact of the correct modelling of the out-of-plane behavior and non-linear connections on the seismic demand parameters, such as base shear and roof displacements, has been highlighted.
The research is of significant interest for researchers involved both in practical and theoretical field of seismic assessment of masonry buildings. This Reviewer only suggests some minor revisions before full acceptance of the paper.
Detailed comments:
1) Pag 4 “ First with-of-plane capability of the microelement and non-linear connections, and then…” please check the sentence; maybe the word “”out” is missing.
2) ) Page 4: please reformulate the sentence “…The wall thickness of both storeys is 60 cm, with the thickness of spandrels under the openings decreased to 30 cm”.
3) Figure 2 is not cited in the text and in the reviewer’s opinion it does not add any useful information. Authors should consider removing it.
4) Page 6: The caption of Table 1 is not clear. What are the values inside the parentheses? How were they used in the analysis? Please specify and describe in more detail.
5) About the selection of the earthquake record, why the Montenegro 1979 earthquake has been selected?
Also the uncertainties of the input motion can influence the response of the structure. Maybe, the Authors could add some general comments on this topic.
6) Page 9: please check and improve the sentence “Among all the limit states, that of collapse is the most challenging to describe, which can be attributed to the influence of non-linear material models,….”
7) Please check minor mistakes of the English language
8) Please check the references. In many of them the journal is missing and/or there are typos. For example see references [15], [19], [21].
